



# Coupled and decoupled stratocumulus-topped boundary layers: turbulence properties

Jakub L. Nowak[1], Holger Siebert[2], Kai-Erik Szodry[2], and Szymon P. Malinowski[1]

[1]Institute of Geophysics, Faculty of Physics, University of Warsaw, Pasteura 5, 02-293 Warsaw, Poland
[2]Leibniz Institute for Tropospheric Research, Permoserstr. 15, 04318 Leipzig, Germany

**Correspondence:** Jakub L. Nowak (jakub.nowak@fuw.edu.pl)

**Abstract.** We compare turbulence properties in two cases of marine stratocumulus-topped boundary layer, coupled (CP) and decoupled (DCP), using high resolution in situ measurements performed by the helicopter-borne platform ACTOS in the region of Eastern North Atlantic.

Thermodynamically well-mixed CP was characterized by large latent heat flux at the surface and in cloud top region, and
substantially smaller sensible heat flux. Turbulence kinetic energy (TKE) was efficiently generated by buoyancy in the cloud and at the surface, and dissipated with comparable rate across the entire depth. Structure functions and power spectra of velocity fluctuations in inertial range were reasonably consistent with the predictions of Kolmogorov theory. The turbulence was close to isotropic.

In the DCP, decoupling was most obvious in humidity profiles. Heat fluxes and buoyant TKE production at the surface were
similar to the CP. Around the transition level, latent heat flux decreased to zero and TKE was consumed by weak stability. In the cloud top region heat fluxes almost vanished and buoyancy production was significantly smaller than for the CP. TKE dissipation rate inside the DCP differed between its sublayers. Structure functions and power spectra in inertial range deviated from Kolmogorov scaling. This was more pronounced in the cloud and subcloud layer in comparison to the surface mixed layer. The turbulence was more anisotropic than in the CP, with horizontal fluctuations dominating. The degree of anisotropy
was largest in the cloud and subcloud layer of the DCP.

Integral lengthscales, of the order of $100\,\mathrm{m}$ in both cases, indicate turbulent eddies smaller than the depth of the CP or of the sublayers of the DCP. We hypothesize that turbulence produced in the cloud or close to the surface is redistributed across the entire CP but rather only inside the relevant sublayers in the DCP. Scattered cumulus convection may play a role in transport between those sublayers.

## 1 Introduction

Low-level stratocumulus clouds (SCs) cover around $20\,\%$ of the Earth's surface in annual mean, more than any other cloud type. They occupy upper few hundred meters of the planetary boundary layer (BL), preferably in the conditions of large-scale subsidence, strong lower-tropospheric stability and moisture supply from the surface (Wood, 2012). Those are usually present in the regions of subtropical and midlatitude oceans with upwelling of cold deep water. Wide-spread presence, persistence and





high albedo makes marine stratocumulus important for the energy balance of the planet (Hartmann et al., 1992). Minor variations in coverage and optical thickness impact the radiation budget, therefore also model-based climate predictions (Boucher et al., 2013; Schneider et al., 2019).

Primary mechanism driving the circulation inside stratocumulus-topped boundary layer (STBL) is longwave radiative cooling at the cloud top which produces convective instability. Additional source of turbulence is provided by surface buoyancy,
wind shear, latent heat release in updrafts, evaporation in downdrafts or evaporative cooling associated with entrainment of dry, warm air from the free troposphere (Lilly, 1968; Stevens, 2002; Gerber et al., 2016; Mellado, 2017). Properties of the STBL are dependent on the level in which SC is coupled with sea surface fluxes, in particular of latent and sensible heat (Bretherton and Wyant, 1997; Xiao et al., 2011; Zheng et al., 2018a).

Moderately shallow STBLs are often well mixed (Stull, 1988; Markowski and Richardson, 2010). Their typical vertical
structure features adiabatic lapse rate (dry below cloud, moist inside), strong capping inversion at the top, near-constant concentration of moist-conserved variables (such as total water mass fraction and liquid water potential temperature) from the surface up to the inversion. However, when the circulation ceases to mix the air over entire depth, the STBL becomes decoupled, i.e. the cloud is separated from the moisture supply from the surface (Nicholls, 1984; Turton and Nicholls, 1987; Wood, 2012). Radiatively driven SC-containing layer (SCL) in the upper part might be still mixed by negatively buoyant eddies gen-
erated at cloud top while the surface mixed layer (SML) at the bottom by positive buoyancy or shear. Stable or conditionally unstable intermediate transition layer (TSL) emerges in between. Conditional instability allows for the cumulus updrafts to penetrate through and intermittently restore the coupling (Bretherton and Wyant, 1997; De Roode and Duynkerke, 1997).

Decoupling can be caused either by reducing the intensity of radiatively driven circulation in relation to STBL depth or by stabilizing the subcloud layer (Zheng et al., 2018b). The first possibility might be realized with daytime shortwave radiative
heating which offsets longwave cooling (Nicholls, 1984; Turton and Nicholls, 1987) or by extensive entrainment of warm and dry free-troposheric air which deepens the STBL to such an extent that the turbulence is no longer sufficient to sustain the mixing (Bretherton and Wyant, 1997). The second possibility involves stratification of the lower part by cooling, for instance due to precipitation evaporation (Caldwell et al., 2005; Dodson and Small Griswold, 2021) or advection over colder sea surface (Stevens et al., 1998).

SC decoupling is the factor which strongly influences further evolution of cloud pattern and boundary layer structure. It constitutes an intermediate stage of transition from overcast stratocumulus into shallow cumulus convection over subtropical oceans as the air masses are advected by the trade winds towards the equator (Albrecht et al., 1995; Bretherton and Wyant, 1997; De Roode et al., 2016; Zheng et al., 2020). Successful representation and prediction of such transition between the two STBL regimes pose a challenge for atmospheric general circulation models (Xiao et al., 2012), in large part due to limited
understanding of the interaction of various processes involved.

Previous observational studies have documented the structure of the coupled and decoupled STBLs in terms of thermodynamic and radiative features (Wood and Bretherton, 2004; Jones et al., 2011; Ghate et al., 2015; Zheng and Li, 2019) as well as aerosol and cloud properties (Dong et al., 2015; Wang et al., 2016; Goren et al., 2018; Zheng et al., 2018b). On the other





hand, modeling efforts provided insightful conceptual explanations of the mechanisms leading to a switch between coupled
and decoupled regimes (Turton and Nicholls, 1987; Bretherton and Wyant, 1997; Stevens, 2000; Xiao et al., 2011).

Although the concept of circulation and turbulence being insufficiently strong in order to maintain the mixing throughout
the entire depth plays a central role in conventional rationale of decoupling, few works attempted to quantitatively characterize
small-scale (integral lengthscales and below) turbulence (e.g. Lambert and Durand, 1999; Dodson and Small Griswold, 2021).
The major reason is the technical difficulty in measuring turbulent fluctuations of wind velocity, temperature or humidity with
adequate spatial resolution and accuracy. Within the present study, we compare the properties of turbulence derived from unique
helicopter-borne observations performed in coupled and decoupled STBL in the region of Eastern North Atlantic. Particular
attention is given to small-scale features and deviations from the assumption of stationary homogeneous isotropic turbulence.

The paper is structured as follows. Section 2 introduces the measurements, including instrumentation, sampling strategy and
general synoptic conditions. The selection of the two cases, coupled and decoupled STBL, is explained. Section 3 describes
the stratification of the STBL in terms of thermodynamics and stability. The division into sublayers is delineated and the
degree of coupling is expressed quantitatively according to literature criteria. Section 4 provides relevant details concerning
derivation of turbulence parameters. Section 5 compares properties of turbulence: turbulence kinetic energy, its production and
dissipation rates, fluxes of sensible and latent heat, anisotropy of turbulent motions, typical lengthscales. Finally, the results of
the comparison are summarized and discussed in the last section.

## 2   Measurements

### 2.1   Location and synoptic conditions

Observations were collected in July 2017 during the ACORES (Azores stratoCumulus measurements Of Radiation, turbulEnce
and aeroSols) campaign in the Eastern North Atlantic (ENA) around the island of Graciosa in the Azores archipelago. Com-
prehensive description of the project, including weather conditions, instrumentation, sampling strategy and selected research
highlights is provided by Siebert et al. (2021).

The area of the experiment is considered to be influenced by a wide range of synoptic scale meteorological conditions.
Graciosa is located near the boundary of subtropics and mid-latitudes. Therefore, the impacts of both subtropical trade wind
system and mid-latitude cyclones are relevant. The climatology of the marine boundary layer was inferred by Rémillard et al.
(2012) based on the long-term ground-based measurements of CAP-MBL project (Wood et al., 2015) utilizing the Atmospheric
Radiation Measurement (ARM) facility established right next to the Graciosa airport. They reported that BL decoupling and
multiple cloud types (for instance cumulus under stratocumulus) are very frequent at the site throughout the year. Indeed, the
range of weather conditions was observed during the ACORES, related to the location and strength of Azores high, as well as
occasional front passages (Siebert et al., 2021).



## 2.2 Instrumentation

Measurements were performed with the Airborne Cloud Turbulence Observations System (ACTOS, Siebert et al. (2006a)) and the Spectral Modular Airborne Radiation measurement sysTem - HELIcopter-borne ObservationS (SMART-HELIOS, Werner et al. (2013, 2014)). Both instrumental payloads were carried by the helicopter BO-105 as two separate external cargos on one long tether: SMART-HELIOS mounted 20 m below the helicopter and ACTOS another 150 m underneath. Typical true air speed (TAS) of 20 m s$^{-1}$ and high sampling rate of individual instruments provided spatial resolution much higher than for a 95 typical research aircraft.

For complete instrumentation of the helicopter payloads see Tables 1 and 2 in Siebert et al. (2021). In the current study we used the following ACTOS data: three-dimensional wind vector $(u_e, v_e, w_e)$ in the Earth-fixed system and longitudinal-vertical wind components $(u, w)$ in platform-fixed system (derivation explained in sec. 4) provided by the combination of the ultrasonic anemometer (Gill Solent HS) and a high-accuracy motion package (inertial navigation system and GPS); temperature $T$ and its 100 small scale fluctuations measured by the Ultra-Fast Thermometer (Haman et al., 1997; Nowak et al., 2018) combined with the precise calibrated PT100; specific humidity $q_v$ from the infrared absorption hygrometer (LICOR 7500, Lampert et al. (2018)); liquid water content $q_l$ determined with the Particle Volume Meter (PVM-100A, Gerber et al. (1994); Wendisch et al. (2002)).

## 2.3 Data overview

Helicopter flights during ACORES were typically performed over the ocean inside the 10 by 10 km square adjacent to the 105 northern coast of Graciosa. Specific flight path and maneuvers depended on local cloud situation. Within the flight time of two hours, usual strategy involved: vertical profile up to roughly 2000 m (a.s.l.), a few 10 km long horizontal legs at selected levels and several steep porpoise dives around SC top. Two flights were selected for our comparative study: flight #5 on 8 July 2017 and flight #14 on 18 July 2017. The choice was dictated by SC presence, STBL stratification (considerably well-mixed in flight #5, considerably decoupled in flight #14) and flight pattern involving substantial sampling time below SC.

Segments of two types were selected from of the measurement records: profiles (PROFs) and horizontal legs (LEGs). For convenience, they are ordered according to their time of execution and referred to as PROF1-PROF5 and LEG1-LEG5, for each flight. The segmentation was done manually so that the influence of sharp turns and pendulum-like motion of the payload is minor. PROFs are in fact slanted with an ascent or descent rate of about 3-5 m s$^{-1}$ and TAS $\sim$20 m s$^{-1}$. The horizontal component of motion is necessary to avoid the downwash of the helicopter affecting wind and turbulence measurements on 115 ACTOS. LEGs were flown with TAS of 15-20 m s$^{-1}$ and some minor displacements in vertical are unavoidable for the payload on a 170 m long rope.

Flight #5 was performed in the afternoon (14:28-16:26 UTC[1]) on 8 July 2017. Stratocumulus clouds emerged behind the cold front which had passed the island the day before. The cloud field was moderately thick and quite heterogeneous in structure, with some visible clearings. Satellite image from MODIS on Aqua (Fig. 1) confirms this observation showing dispersed cloud

---

[1]On Azores the local time in summer is equivalent to UTC.





patches in the vicinity of Graciosa . The flight pattern (Fig. 2) involved: deep PROF from minimum flight level (60 m) into free
troposphere (FT), two LEGs in the FT with one close to SC top, three LEGs in the STBL with one inside SC, close to its top.

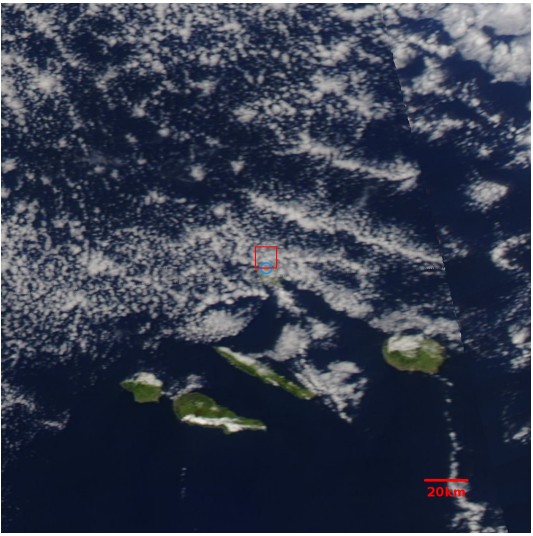

**Figure 1.** Satellite true color image (250 x 250 km) from MODIS on Aqua overpassing Azores during flight #5, centered on Graciosa airport
(blue circle), with overlaid helicopter operation area (red box). The image was acquired from NASA Worldview Snapshots.

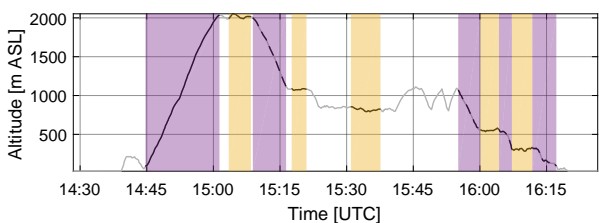

**Figure 2.** ACTOS altitude in flight #5 with marked selected profiles (purple) and horizontal legs (yellow). Line style of the profiles is
consistent with the figures in following sections.

Flight #14 was performed in the afternoon (15:01-17:04 UTC) on 18 July 2017, shortly after weak precipitation had been
noted at the site. The sky was overcast with SC of homogeneous structure. Many little cumulus clouds, probably at the initial
state of formation, were reported over the ocean below SC deck. However, they were not observed to reach SC base. MODIS
Aqua image (Fig. 3) shows large solid patch of SC with signatures of closed-cell convection regime. The flight pattern (Fig. 4)
involved: four LEGs in the STBL with one inside SC, close to its top, one LEG in the FT and a number of PROFs connecting
LEG levels.





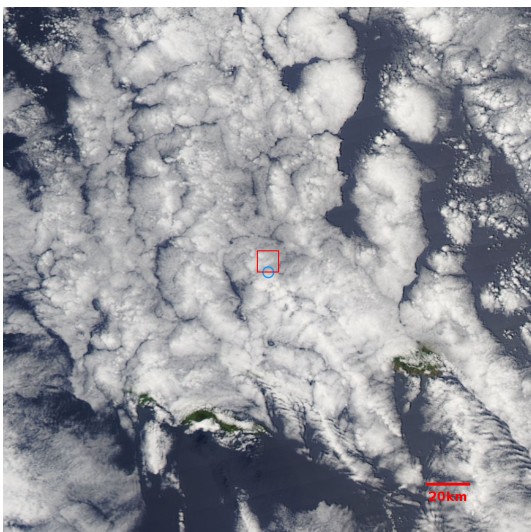

**Figure 3.** As in Fig. 1 but for flight #14. The image was acquired from NASA Worldview Snapshots.

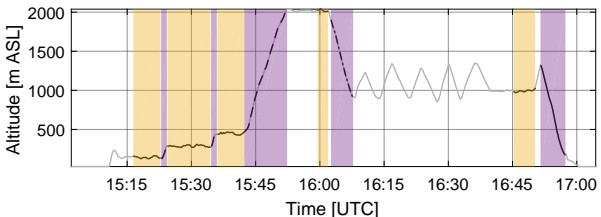

**Figure 4.** As in Fig. 2 but for flight #14.

# 3 Stratification

## 3.1 Derivation of meteorological and stability parameters

Meteorological conditions and stability parameters derived from PROFs are shown in Fig. 5 and Fig. 6 for flight #5 and #14, respectively. Liquid water potential temperature $\theta_l$ was calculated following the approximation by Betts (1973):

$$\theta_l = \theta - \frac{\theta}{T}\frac{L_v}{c_p}q_l \tag{1}$$

where $\theta$ denotes potential temperature, $L_v$ latent heat of vaporization for water and $c_p$ specific heat of dry air at constant pressure. Horizontal wind speed $U$ and direction $dd$ result from appropriate transformation of measured flow velocity (Edson

et al., 1998). Because helicopter climb rate was not exactly constant in time and individual instruments differ in sampling rate, data points were grouped and averaged in 10 m high altitude bins (yet separately for each PROF). To reduce the effect of random eddy penetration and improve clarity, wind profiles were additionally smoothed with five point moving average.



Lifting condensation level (LCL) was then derived for each height according to Bolton (1980). Such result is sensitive to gradients of thermodynamic properties in subcloud layer, signaling the degree of BL coupling. To characterize static stability, 140 Brunt-Vaisala frequency Nb was used:

$$\mathrm{Nb}^2 = \frac{g}{\theta_v}\frac{\partial \theta_v}{\partial z} \tag{2}$$

where $\theta_v$ is virtual potential temperature derived from speed of sound (provided by ultrasonic anemometer), $g$ gravitational acceleration and $z$ height above sea level. Shear rate Sh quantifies vertical gradient of horizontal wind:

$$\mathrm{Sh}^2 = \left(\frac{\partial u_e}{\partial z}\right)^2 + \left(\frac{\partial v_e}{\partial z}\right)^2 \tag{3}$$

where $u_e$ is eastward and $v_e$ northward wind component. The derivatives were evaluated as the tangent of linear least-square fit of 10 m binned variable versus $z$ performed inside symmetric five point windows.

### 3.2 Quantitative judgement of the degree of coupling

In order to objectively confirm the fact of coupling or decoupling of STBL, we employed several methods from the literature: Jones et al. (2011), hereafter J11, Wood and Bretherton (2004), hereafter WB04, Yin and Albrecht (2000), hereafter YA00.

First criterion of J11 involves the differences of $\theta_l$ and total water content $q_t = q_l + q_v$ between the uppermost and the lowermost quarters of BL. The sounding is classified as coupled when $\Delta\theta_l = \theta_l^{top} - \theta_l^{bot} < 0.5$ K and $\Delta q_t = q_t^{bot} - q_t^{top} < 0.5\,\mathrm{g\,kg^{-1}}$, decoupled otherwise. Second criterion of J11 involves the difference between observed cloud base height (CB) and the LCL corresponding to the conditions at BL bottom. The BL is classified as coupled when $\Delta z = \mathrm{CB} - \mathrm{LCL}^{bot} < 150$ m, decoupled otherwise. Here, we used mean conditions of the lowest leg (LEG5 for flight #5, LEG1 for flight #14) to estimate $\mathrm{LCL}^{bot}$ and 155 $q_l$ in PROFs to estimate CB.

WB04 proposed two decoupling parameters:

$$\alpha_\theta = \frac{\theta_l^- - \theta_l^0}{\theta_l^+ - \theta_l^0} \qquad \alpha_q = \frac{q_t^- - q_t^0}{q_t^+ - q_t^0} \tag{4}$$

where superscripts $+, -, 0$ denote the values just above the inversion, just below the inversion and in the surface mixed layer, respectively. WB04 calculated $\alpha_\theta$ and $\alpha_q$ over subtropical Eastern Pacific at around 0 to 0.4, however no exact critical value 160 for decoupling was determined. The higher those parameters, the more decoupled BL is considered. Here, instead of finding first the SML, we apply mean values in the lower quarter of the BL ($\theta_l^0 = \theta_l^{bot}$ and $q_t^0 = q_t^{bot}$).

YA00 introduced a stability parameter to identify transitions in BL soundings:

$$\mu = -\frac{\partial \theta}{\partial p} + \frac{\varepsilon\theta}{1+\varepsilon r}\frac{\partial r}{\partial p} \tag{5}$$

where $\varepsilon = R_v/R_d - 1$ depends on the ratio of gas constants for water vapor $R_v$ and dry air $R_d$, while $r$ is water vapor mixing 165 ratio. Their procedure detects transition anytime in the subcloud zone the value of $\mu$ exceeds by a factor of 1.3 the average $\bar{\mu}$ between 980 and 900 hPa. Here, instead of using pressure levels, we specify $\bar{\mu}$ as BL mean.



The above parameters were estimated using PROF1 of flight #5 and PROF5 of flight #14. According to J11 criteria, it is evident that flight #5 ($\Delta\theta_l = -0.51°$C, $\Delta q_t = 0.13$ g kg$^{-1}$, $\Delta z = -72$ m) was performed in coupled STBL while flight #14 ($\Delta\theta_l = 1.19°$C, $\Delta q_t = 0.90$ g kg$^{-1}$, $\Delta z = 216$ m) in decoupled STBL. Negative values suggest instability but it might be also attributed to horizontal inhomogeneities of SC structure (sec. 2.3) in combination with slanted flight path. Consistently, WB04 parameters are smaller for flight #5 ($\alpha_\theta = -0.12$, $\alpha_q = 0.04$) than for flight #14 ($\alpha_\theta = 0.26$, $\alpha_q = 0.26$). The parameter of YA00 is plotted in panel (d) of Figs. 5 and 6. It varies significantly with height and the critical value is occasionally exceeded in both flights. This method was probably optimized for radiosoundings in different climate regime and does not seem robust in case of our data.

Following previous studies looking for differences of cloud top entrainment instability (CTEI) between coupled and decoupled SC (e.g. Xiao et al., 2011), we calculated the Randall-Deardorff parameter (Randall, 1980; Deardorff, 1980):

$$\kappa = 1 + \frac{c_p}{L_v} \frac{\theta_l^+ - \theta_l^-}{q_t^+ - q_t^-}. \tag{6}$$

In both our cases ($\kappa = 0.71$ for flight #5 and $\kappa = 0.34$ for #14), it exceeds the critical value of about 0.23 indicating the possibility of buoyancy reversal resulting from mixing and evaporative cooling at cloud top.

## 3.3 Structure of the coupled STBL

The profiles in flight #5 exhibit a well-mixed STBL (Fig. 5). Temperature falls with height with near constant lapse rate $\Gamma_T$ inside the BL, followed by a sharp inversion at the top. Liquid water potential temperature is almost constant from close to the surface up to SC top, where it features the increase of $\sim$5 K. Total water content behaves analogously, with the decrease of $\sim$7 g kg$^{-1}$ above cloud top. Interestingly, very dry air is located at the top of the temperature inversion. It is further capped by the layer of considerably higher $q_v$, however much lower than inside the BL. Liquid water content in the cloud is moderate and suggest non-trivial cloud stucture, probably there were some clearings penetrated as ACTOS moved along the slanted path. Wind velocity fluctuates in the BL within $\pm 1$ m s$^{-1}$ around the mean $\sim$5 m s$^{-1}$. Wind shear across the cloud top and the inversion can be noticed. Wind direction is from the NNE throughout the sampled height.

Significant differences can be observed between the PROFs in wind speed and the position of inversion. Subsequent PROFs were not performed at the same time and location, so certain variability is expected. Airborne sampling features inevitable randomness due to probing specific structures (eddies, updrafts, cloud holes etc.), thus slanted profiles do not represent mean conditions accurately.

LCL stays roughly equal from the lowest level up to the cloud base. Interestingly, it is slightly higher than the actual CB which might be again related to horizontal inhomogeneities in cloud structure. Brunt-Vaisala frequency indicates weak static instability in the BL, stronger inside the cloud than below, and very strong stability at the capping inversion. Wind shear is more variable which can be attributed to sampling various eddies.

Based on $\theta_l$, $q_l$ and $q_v$, we distinguished the following sublayers: the entrainment interface layer (EIL) including the temperature inversion and the very top of the cloud, the stratocumulus layer (SCL) containing the cloud, the subcloud layer (SBL) ranging from cloud base down to the surface, and the sublayer representing free tropospheric conditions (FTL, not necessarily





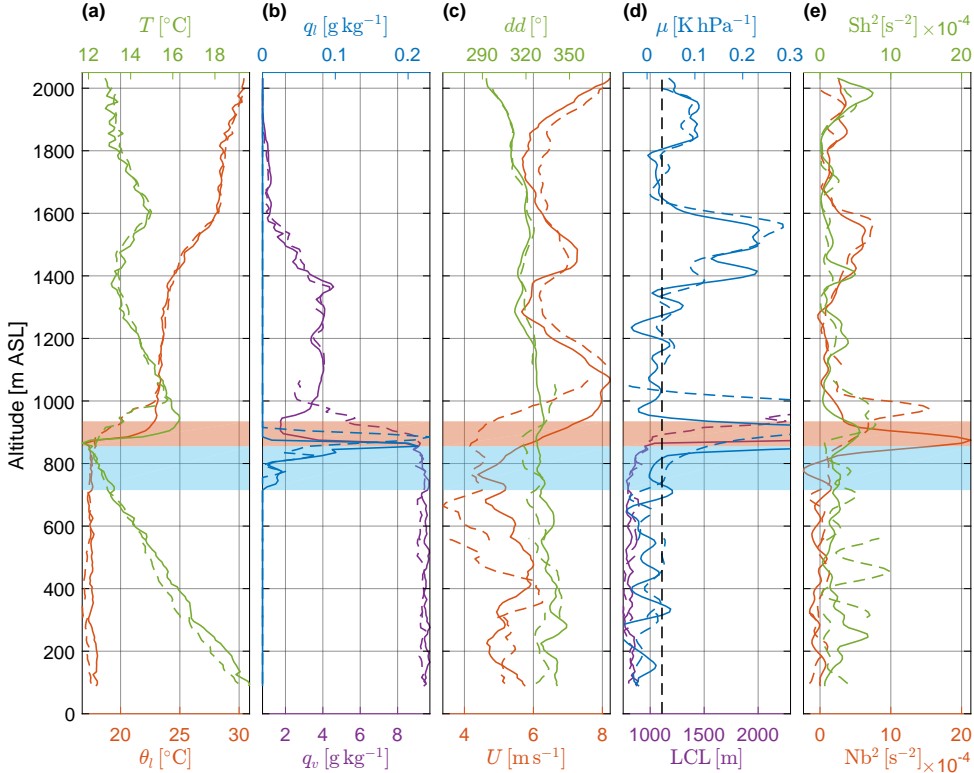

**Figure 5.** Vertical structure of the coupled STBL (flight #5): (a) temperature $T$ and liquid water potential temperature $\theta_l$, (b) liquid water content $q_l$ and specific humidity $q_v$, (c) wind speed $U$ and direction $dd$, (d) lifting condensation level LCL and stability parameter $\mu$ of Yin and Albrecht (2000) with its critical level for the detection of transitions (dotted black line), (e) squared Brunt-Vaisala frequency $\mathrm{Nb}^2$ and wind shear rate $\mathrm{Sh}^2$. Line styles correspond to specific profiles – consistently with Fig. 2. Color shadings denote the sublayers: entrainment interface layer (red) and stratocumulus layer (blue).

adjacent to the EIL top). For reference, the EIL and SCL are marked with red and blue shading in Fig. 5 and following. The heights and average properties inside the sublayers are listed in Table A1 in the appendix. The deepest profile (PROF1, solid line), was used for sublayer distinction because the specific heights may vary between PROFs. Suitable normalization and averaging (Ghate et al., 2015) is not possible in our study because other PROFs are not deep enough.

### 3.4 Structure of the decoupled STBL

The profiles in flight #14 exhibit decoupled STBL (Fig. 6). Liquid water potential temperature gradually rises with height whereas specific humidity decreases step-wise. Despite the distinct $q_v$ gradient in BL middle, its value in the lowest part and in the subcloud section is relatively stable. This suggests the upper BL portion is internally mixed and the lower BL portion is internally mixed. The FT is quite humid, with values of $q_v$ larger than for flight #5. The difference in $\theta_l$ at SC top is $\sim$5 K while



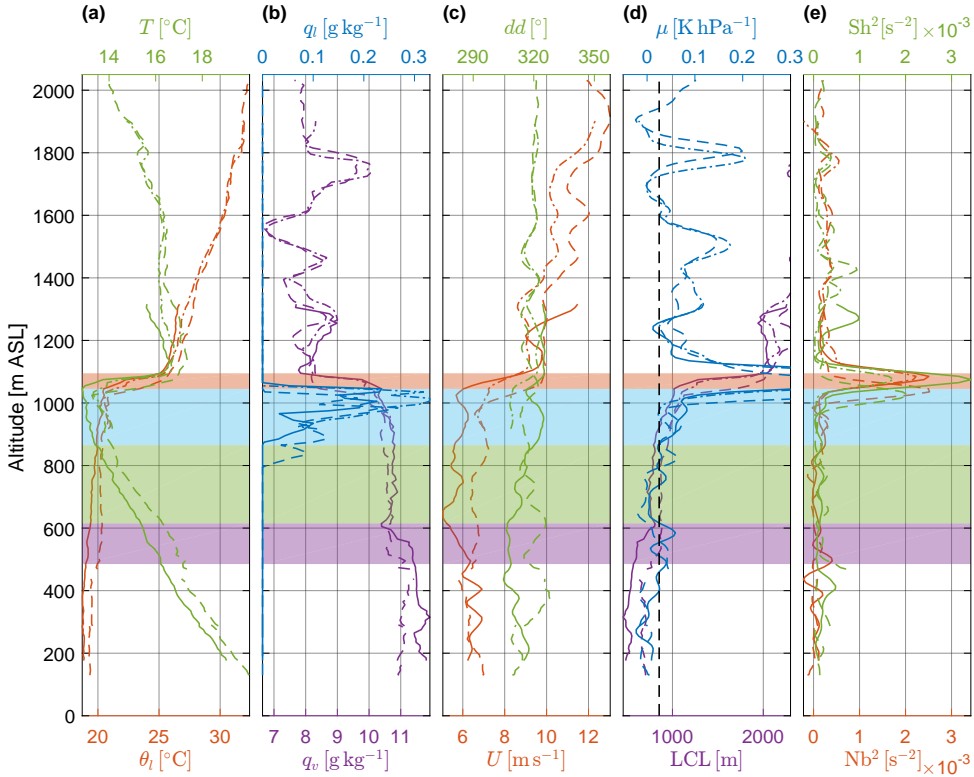

**Figure 6.** As in Fig. 5 but for the decoupled STBL (flight #14). Line styles are consistent with Fig. 4. Color shadings denote the sublayers: entrainment interface layer (red), stratocumulus layer (blue), subcloud layer (green), transition layer (purple).

in $q_t$ only ~3 g kg$^{-1}$. The stratocumulus is thicker and more abundant in liquid water than in the previous case. Wind velocity
varies ±1 m s$^{-1}$ around the mean ~6 m s$^{-1}$. Wind direction is predominantly NE. There is significant wind shear across the inversion, with difference in $U$ reaching ~4 m s$^{-1}$. LCL replicates the gradients of $q_v$ in the middle BL. It corresponds to the CB only in the section right below the cloud which is a signature of decoupling. Brunt-Vaisala frequency indicates weak static stability throughout most of the profile, including the cloud. Its peak in the inversion layer coincides well with the maximum of Sh$^2$.

Similarly to flight #5, we distinguished the sublayers: the FTL, the EIL, the SCL, and the SBL extending from cloud base down to the level where LCL is no longer in agreement with the observed CB. In addition, two more sublayers typical for decoupled conditions were introduced: the transition layer (TSL) containing the major gradients in specific humidity and wind speed, and the surface mixed layer (SML) extending from the surface up to the bend in $\theta_l$ profile (where it begins to rise with height, c.f. Fig. 6a). Somewhat arbitrary boundary of 385 m was chosen to represent the section directly influenced by surface
processes. For reference, the EIL, SCL, SBL and TSL are marked with red, blue, green and purple shading, respectively, in





Fig. 6 and following. The heights and average properties inside the sublayers are listed in Table A2. PROF5 was used for sublayer identification because it covers most of the STBL depth.

## 4 Turbulence properties: methods

Parameters of turbulence were derived using high-resolution measurements of wind velocity, temperature and humidity. De-
pending on the quantity, the results were obtained for PROFs or LEGs specified in sec. 2.3. In case of PROFs, our procedure resembles the approach of Tjernstrom (1993). After timeseries of a parameter had been computed, appropriate segments were extracted and data were averaged in 10 m altitude bins (as in sec. 3.1). For LEGs, full segment was used to calculate a desired parameter. Next, each LEG was divided into 7 subsegments of equal length, overlapping by half of the length, and the very same method was applied to calculate respective quantity in each subsegment. Standard deviation among subsegments is
regarded as parameter variability and shown with errorbars in plots.

The lateral channel of the ultrasonic anemometer was affected by a substantial level of artificial fluctuations (up to $1\,\mathrm{m\,s^{-1}}$ in amplitude) due to instrumental issues. Therefore, we applied simplified geometrical transformation to the measured velocity vector, so that high resolution retrieval of wind velocity is possible. In comparison with the standard transformation (Lenschow, 1986), we included pitch rotation but neglected roll and yaw rotations to prevent the lateral channel from coupling with the
others. The resulting vector $(u, v, w)$ can be interpreted as wind velocity in horizontal longitudinal, horizontal lateral and vertical direction, respectively, as long as the platform is not tilted left or right (roll angle is small). This condition was satisfied throughout most of the flight time, except for major turns. For calculating turbulence properties, we selected segments with the roll angle $<0.1\,\mathrm{rad}$. The lateral wind $v$ cannot be used for turbulence analysis but longitudinal $u$ and vertical $w$ are free from the disturbances. Described modification is not necessary to obtain mean wind profiles ($U$, $dd$) because averaging and
smoothing is applied anyway (see sec. 3.1).

Reynolds decomposition of the signals (c.f Stull, 1988)

$$x(t) = X(t) + x'(t) \tag{7}$$

into large scale slowly varying $X(t)$ and small scale fluctuations $x'(t)$ was realized with simple symmetric running mean. Fluctuations $x'(t)$ were obtained by subtracting that mean from original signal. Unless specified otherwise, the chosen window
was $50\,\mathrm{s}$ which corresponds to the distance of $\sim 1\,\mathrm{km}$. Such length is enough to penetrate at least a few large turbulent eddies typical for the atmospheric boundary layer (Malinowski et al., 2013).

### 4.1 Turbulence Kinetic Energy and variances

Variances of turbulent fluctuations $\langle u'^2 \rangle$, $\langle w'^2 \rangle$, $\langle T'^2 \rangle$, $\langle q_v'^2 \rangle$ and third moment of vertical velocity fluctuations $\langle w'^3 \rangle$ were obtained by taking average along LEG, denoted as $\langle \rangle$. Because lateral wind fluctuations were not available, we assumed
horizontal isotropy to approximate missing $\langle v'^2 \rangle$ with $\langle u'^2 \rangle$ in turbulence kinetic energy (TKE) calculation:

$$\mathrm{TKE} = \langle u'^2 \rangle + \frac{1}{2}\langle w'^2 \rangle. \tag{8}$$





Worth to remember, variances and TKE usually represent mostly large scales because larger eddies in turbulence cascade are more energetic than smaller ones.

## 4.2 TKE production and heat fluxes

Turbulence kinetic energy can be generated by buoyancy and wind shear (ignoring advection and turbulent transport). We estimated two respective terms of the TKE budget equation (Stull, 1988), buoyancy production/consumption $B$ and shear production $S$, employing eddy correlation:

$$B = \frac{g}{\langle \theta_v \rangle} \langle w' \theta_v' \rangle, \qquad S = -\langle w'u' \rangle \frac{\partial u}{\partial z}. \tag{9}$$

Here, we could provide only longitudinal component of shear production because lateral wind fluctuations were not available.
Correlations were computed along the LEGs. Derivatives were estimated from the PROFs covering the relevant altitude range. Inevitably, such approach introduces some inaccuracy as the exact place and time of derivative estimation is different than for the correlation estimation. To quantify vertical transport of heat and moisture, we estimated sensible and latent heat fluxes according to:

$$Q_s = \rho c_p \langle w' \theta' \rangle, \qquad Q_l = \rho L_v \langle w' q_v' \rangle \tag{10}$$

where $\rho$ is air density.

Range of scales represented in the correlations is limited by the smaller among spatial resolutions of two multiplied signals. The anemometer ($u$, $w$, $\theta_v$) resolves scales down to $\sim$0.5 m (where this limit stems from the path length and spectral transfer properties (Kaimal et al., 1968)), the thermometer ($\theta$) down to $\sim$2 cm, the hygrometer ($q_v$) down to $\sim$1 m. Those three instruments work satisfactorily also inside clouds of moderate liquid water and droplet concentration, as our SC (Cruette
et al., 2000; Siebert and Teichmann, 2000). In comparison with some other studies, the buoyancy estimation in the cloud does not include the contributions of liquid water flux $\langle w' q_l' \rangle$ and droplet sedimentation which are expected to be relatively small (considering moderate $q_l$) and of opposite sign, therefore partly compensate.

Additionally, $B$, $Q_s$ and $q_l$ at the surface were estimated with the Coupled Ocean–Atmosphere Response Experiment bulk algorithm in version 3.0 (COARE 3.0) described in Fairall et al. (2003). Sea surface temperature was taken from satellite
multi-mission product provided by the Group for High Resolution Sea Surface Temperature (JPL MUR MEaSUREs Project, 2015) while all the other inputs were our measurements from the lowest point of the PROFs.

## 4.3 TKE dissipation rate

TKE dissipation rate $\epsilon$ was calculated invoking common assumption of homogeneous, isotropic, stationary turbulence which leads to the specific form of power spectra and structure functions (Kolmogorov, 1941). Nevertheless, theoretical assumptions
are often hardly satisfied in the atmosphere, e.g. considering complex stratification, and therefore $\epsilon$ estimation from moderate-resolution (not directly resolving dissipative scales) measurements is challenging (Siebert et al., 2006b; Jen-La Plante et al., 2016; Wacławczyk et al., 2017, 2020). To account for possible anisotropy, $\epsilon$ was derived separately for longitudinal and vertical





velocity fluctuations, following the methods of Siebert et al. (2006b). We also characterized the quality of estimations with additional parameters describing the deviation of experimental data from theoretical dependencies.

### 285    4.3.1    Structure function method

Second order structure function (SFCs) was calculated for measured $u'$ and $w'$ according to the same equation:

$$D_u(r) = \left\langle |u'(x+r) - u'(x)|^2 \right\rangle \tag{11}$$

where $r$ is distance between data points (given by TAS) and the average is taken over positions $x$ along the flight path. SFC was then resampled, i.e. averaged inside logarithmically equidistant bins covering the assumed inertial range $r \in [0.4, 40]$ m,
with eight bins per decade (see Fig. 7). The resampling was applied in order to account for the density of data points increasing with scale in logarithmic coordinates.

Theory predicts that in inertial range SFC has the form (Pope, 2000):

$$D(r) = C(\epsilon r)^{\frac{2}{3}} \tag{12}$$

where $C$ is a constant, experimentally determined to $C_u \approx 2.0$ for longitudinal and $C_w \approx 2.6$ for lateral velocity component.
We calculated $\epsilon^{sfc}$ by least squares fit of this relationship to the resampled SFC. Second fit was perfomed according to:

$$D(r) = C^* r^s \tag{13}$$

with two fitted parameters: prefactor $C^*$ and exponent $s$ corresponding to the slope in log-log plot. The exponent is used as a benchmark of the agreement of the SFC form with theory. Additionally, Pearson correlation coefficient $R^{sfc}$ was computed for the resampled points. It quantifies the linearity of the experimental SFC in log-log coordinates. Consequently, $s$ and $R^{sfc}$
assess to some extent the reliability of derived $\epsilon$.

### 4.3.2    Power spectrum method

Power spectral density (PSD) of $u'$ and $w'$ was calculated with the Welch algorithm. The window was chosen as half the length of the segment. The windows overlap by half of their length, so in turn there are three individual PSDs averaged in the Welch scheme. PSD was resampled in the assumed inertial range, analogously to SFC (see Fig. 8).
Theory predicts the following PSD form in inertial range (Pope, 2000):

$$P(f) = C' \left( \frac{U_s}{2\pi} \right)^{\frac{2}{3}} \epsilon^{\frac{2}{3}} f^{-\frac{5}{3}} \tag{14}$$

where $f$ is frequency and $C'$ is a constant ($C'_u \approx 0.49$ for longitudinal and $C'_w \approx 0.65$ for lateral component). We derived $\epsilon^{psd}$ by fitting this relationship to the resampled PSD. Second fit was performed according to:

$$P(f) = C^* f^p \tag{15}$$

where fitted PSD exponent $p$ corresponds to the slope in log-log plot. Together with Pearson correlation coefficient for the resampled points $R^{psd}$ it measures the agreement of PSD form with theory and reliability of derived $\epsilon$.





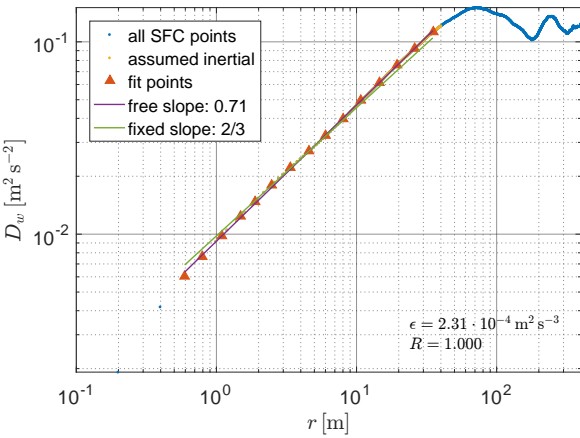

**Figure 7.** Example of $\epsilon$ derivation with structure function method (flight #5, LEG5, vertical component). Computed SFC (Eq. (11), blue) is resampled in the assumed inertial range (yellow) to obtain logarithmically spaced points (triangles) which are used for least squares fits: one with free slope (Eq. (13), purple), one with fixed theoretical slope (Eq. (12), green).

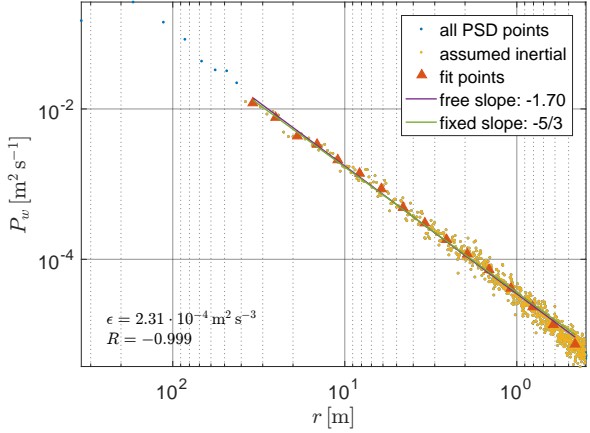

**Figure 8.** Example of $\epsilon$ derivation with power spectrum method (flight #5, LEG5, vertical component). Computed PSD (blue) is resampled in the assumed inertial range (yellow) to obtain logarithmically spaced points (triangles) which are used for least squares fits: one with free slope (Eq. (15), purple), one with fixed theoretical slope (Eq. (14), green).

### 4.3.3 Application of the methods

For PROFs, the moving window of 2 s was applied to the timeseries $u'$ and $w'$. In each window, $\epsilon$ was derived separately with the two methods, together with $s$, $R^{sfc}$, $p$, $R^{psd}$. Such a solution was verified to provide sufficiently good fits and constitutes





the compromise between high final spatial resolution (short window desired) and adequate representation of SFC or PSD (long
window desired).

In case of LEGs, both methods were applied to the whole segment. Then, SFC and PSD were in practice averaged over
relatively long horizontal distance. This approach provides an estimate of mean dissipation in contrast to local values computed
in short windows which might differ from the mean (Kolmogorov, 1962). Also, SFC and PSD derived on long horizontal
segment are expected to follow the theoretical form more accurately which is indeed the case.

Our results (sec. 5.3) demonstrate a good agreement between the methods as long as relative variations with height are
concerned. In terms of absolute values, $\epsilon^{psd}$ is usually higher than $\epsilon^{sfc}$ (around the factor of 2). In general, derived SFC
resembles its theoretical form better than PSD which is indicated by the fitted exponents and correlation coefficients. This
agrees with Siebert et al. (2006b) who found the SFC method to be more robust for $\epsilon$ estimation from airborne platforms.

### 4.4 Anisotropy

The assumption of isotropy might be violated in many specific situations in the atmospheric boundary layer, e.g. under strong
buoyancy and wind shear at SC top (Malinowski et al., 2013; Jen-La Plante et al., 2016; Akinlabi et al., 2019). To investigate
deviations from isotropy, we use anisotropy ratios $A$ of two types, bulk and spectral, relating $w$-derived parameters to $u$-derived
ones.

We define the following bulk anisotropy ratios:

$$A_2^{var} = \sqrt{\frac{\langle w'^2 \rangle}{\langle u'^2 \rangle}}, \qquad A_\epsilon^{sfc} = \frac{\epsilon_w^{sfc}}{\epsilon_u^{sfc}}, \qquad A_\epsilon^{psd} = \frac{\epsilon_w^{psd}}{\epsilon_u^{psd}}. \qquad (16)$$

The first relates mostly to larger eddies which have dominant contribution to total variance. Isotropy is indicated by the values
close to 1, while $A_2^{var} < 1$ and $A_2^{var} > 1$ indicate anisotropic turbulence dominated by horizontal and vertical fluctuations,
respectively. On the other hand, $A_\epsilon^{sfc}$ and $A_\epsilon^{psd}$ regard mostly inertial range eddies because $\epsilon$ derivation exploits SFC or PSD
scaling in inertial range. Analogously, values close to unity indicate isotropy.

The spectral anisotropy is the scale-dependent ratio of PSDs for vertical and longitudinal velocity:

$$A_P(r) = \frac{P_w(U_s/r)}{P_u(U_s/r)} \qquad (17)$$

where TAS is utilized to convert frequency into distance. Similar approach was exercised by Pedersen et al. (2018) who
compared modeled and measured anisotropy in the region of SC top. In inertial range, Kolmogorov theory predicts $A_P = 4/3$.
Such value of experimentally derived $A_P(r)$ should then indicate isotropy at the particular scale $r$. We applied the same
resampling procedure as in sec. 4.3.2 to LEG-derived PSDs but across the whole available range of scales (not only inertial)
and the ratio was then calculated point-by-point.

### 4.5 Lengthscales

Turbulence energy cascade is often characterized by several lengthscales: integral scale $L$, Taylor microscale $\lambda$ and Kolmogorov
scale $\eta$. Integral lengthscale corresponds to energy-containing eddies which are involved in TKE generation. In energy cascade,





it marks the beginning of inertial subrange where turbulent flow is considerably isotropic despite the anisotropy of large scale factors. The indefinite integral of autocorrelation function involved in formal definition of $L$ cannot be evaluated experimentally due to limited length available. We estimated the distance where the autocorrelation

$$\rho_u(r) = \frac{\langle u'(x+r)u'(x) \rangle}{\langle u'^2 \rangle} \tag{18}$$

declines by a factor of $e$. This method is robust enough to provide reasonable results in all our cases. The very same procedure was applied to longitudinal as well as vertical velocity to provide $L_u$ and $L_w$, respectively. According to Pope (2000), under isotropic conditions $L_w = \frac{1}{2}L_u$. Such proportion can then indicate isotropy in relevant large eddy scale.

At Taylor microscale, viscosity starts to substantially affect the dynamics of turbulent eddies. Under assumption of isotropy, it can be related to velocity variance and dissipation rate. We estimated two Taylor scales, longitudinal and vertical:

$$\lambda_u = \sqrt{30\nu \frac{\langle u'^2 \rangle}{\epsilon_u^{sfc}}}, \qquad \lambda_w = \sqrt{15\nu \frac{\langle w'^2 \rangle}{\epsilon_w^{sfc}}} \tag{19}$$

where $\nu$ is air viscosity for which we accounted for temperature and pressure dependence (Sutherland, 1893). In homogeneous isotropic turbulence $\lambda_w = \frac{1}{\sqrt{2}}\lambda_u$ (Pope, 2000).

Kolmogorov scale corresponds to smallest eddies where TKE is dissipated into heat by viscosity. Following dimensional arguments of the famous similarity hypothesis, it equals:

$$\eta_u = \left( \frac{\nu^3}{\epsilon_u^{sfc}} \right)^{\frac{1}{4}}. \tag{20}$$

It was calculated separately for longitudinal ($\eta_u$) and vertical ($\eta_w$) direction with the same formula. Provided local small-scale isotropy, they should be equal. For convenience, in $\lambda$ and $\eta$ derivation, we used only $\epsilon^{sfc}$ and neglected $\epsilon^{psd}$ because SFC proved to resemble its theoretical form better (see sec. 5.3).

## 5   Turbulence properties: results

Turbulence properties in coupled and decoupled STBL are documented in a series of plots. Depending on flight segment type, they are illustrated with continuous profiles (PROF) and/or dots with errorbars (LEG). For reference, the figures include the sublayer shading introduced in sec. 3. Mean PROF-derived values inside the sublayers are listed in Tables A1 and A2.

### 5.1   Turbulence Kinetic Energy and variances

Figs. 9 and 10 present variances of vertical and longitudinal velocity fluctuations, TKEs, third moments of vertical velocity,
variances of temperature and specific humidity in the LEGs of flight #5 and #14, respectively. Generally, the TKE inside the coupled STBL decreases with height from the middle of the SBL up to the cloud top. Despite slightly unstable stratification, the contribution from horizontal velocity variance is dominant over the vertical one. The latter reaches minimum value below the cloud, where the buoyancy production is close to zero (compare Fig.11 in the next section).





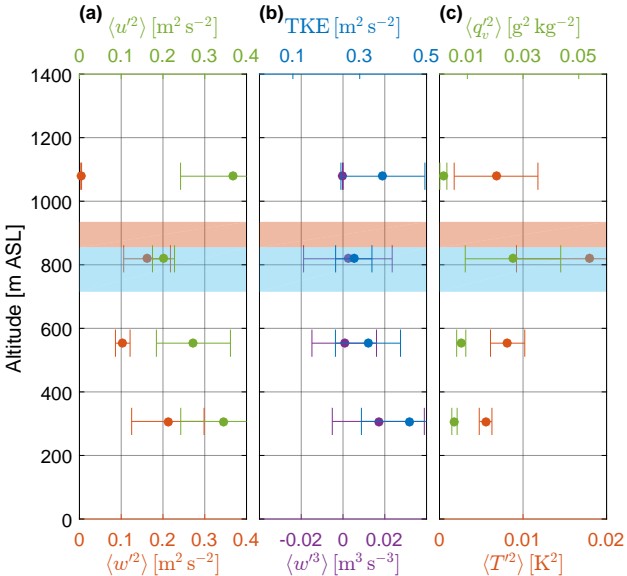

**Figure 9.** Statistics of turbulent fluctuations in the coupled STBL (flight #5): (a) variance of horizontal $\langle u'^2 \rangle$ and vertical velocity $\langle w'^2 \rangle$, (b) turbulence kinetic energy TKE and third moment of vertical velocity $\langle w'^3 \rangle$, (c) variance of temperature $\langle T'^2 \rangle$ and specific humidity $\langle q_v'^2 \rangle$.

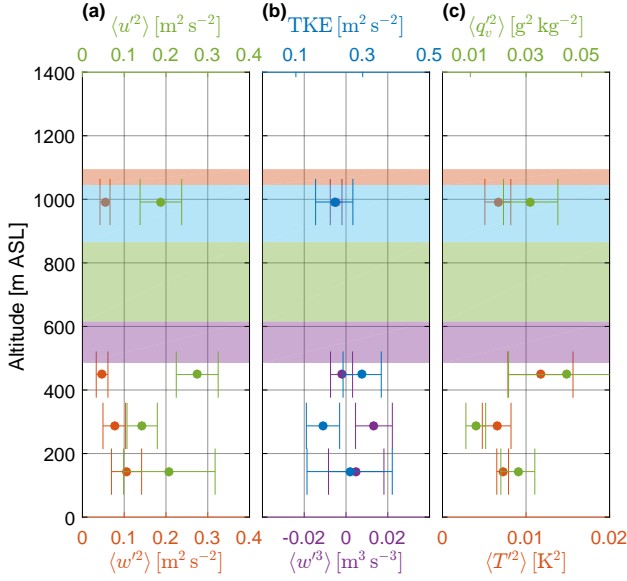

**Figure 10.** As in Fig. 9 but for the decoupled STBL (fight #14).

Estimated values of the TKE are also large in the FT above the temperature inversion. This is rather an artifact due to likely
presence of gravity waves favored under stable conditions. Recall that LEG2 was flown very close to the EIL and the cloud





top which often features undulated interface. Structure function, autocorrelation function as well as simple inspection of the velocity signal indicate oscillations of the wavelength of a few hundred meters.

The third moment of vertical velocity is positive in the lowest LEG at about 300 m altitude, suggesting strong but localized updrafts and weak but widespread downdrafts. Higher up, it is close to zero. This results ought to be interpreted with caution
because the estimation of $\langle w'^3 \rangle$ can be subject to errors due to insufficient statistics related to the small chance of penetrating infrequent but intense events (Lenschow et al., 1994; Kopec et al., 2016).

Fluctuations of temperature and humidity can be significant wherever there are spatial gradients of those quantities or in the presence of sources or sinks of heat and moisture. Such conditions occur close to the cloud top, where radiative cooling is the sink of heat and mixing between the air volumes of considerably contrasting properties occurs. Indeed, measured variances are
highest in the cloud segment and decrease downward into the boundary layer where $T$ and $q_v$ are locally more uniform.

In the decoupled STBL, TKE level is in general lower than in the coupled case. The profiles of velocity variances across the SML resemble typical mixed layer with shear, i.e. high TKE at the bottom and the top which is realized mostly by the contribution of horizontal velocity variance (e.g., Stull, 1988, ch. 4). The prevalence of horizontal in comparison to vertical is particularly visible for LEG3, close to the transition, where the vertical velocity variance reaches its minimum. Similarly to
TKE, humidity and temperature variances exhibit maximum at this level. $T$ and $q_v$ can be considered passive scalars with no significant sources there. The TSL features the gradient of $q_v$ (c.f. Fig. 6) which might explain increased local fluctuations.

Skewness of vertical velocity is slightly positive in the SML with the maximum in LEG2. At the transition and in the cloud it is close to zero with tendency towards negative values. This suggests dominant role of updrafts in the SML and downdrafts in the SCL. Altogether, the results can be interpreted as a signature of decoupling between the circulations in lower and upper
parts of the boundary layer, as downdrafts originated at cloud top and updrafts originated at the surface seem to slow down and diverge horizontally at the transition level.

## 5.2  TKE production and turbulent fluxes

Buoyant production of TKE is expected to be significant inside the cloud and close to the surface while the shear production at the bottom and at the top of the boundary layer (Markowski and Richardson, 2010). Such a picture is in general agreement
with our results for flight #5. In the coupled STBL observed there (Fig. 11), $B$ is maximum in the LEG flown inside the cloud ($8.0 \cdot 10^{-4}\ \mathrm{m^2\,s^{-3}}$), drops to nearly zero below the cloud and increases towards the surface, reaching $5.6 \cdot 10^{-4}\ \mathrm{m^2\,s^{-3}}$ (estimated with COARE algorithm). $S$ appears to be more uniform in the boundary layer, yet subject to substantial variability among subsegments.

Sensible heat flux reaches maximum of almost 40 $\mathrm{W\,m^{-2}}$ close to the cloud top, stays small and positive in the middle of
the boundary layer with the surface value of around $Q_s = 11\ \mathrm{W\,m^{-2}}$ (according to COARE parameterization). Latent heat flux seems to follow near linear decrease from $Q_l = 130\ \mathrm{W\,m^{-2}}$ at the ocean surface, which is the source of moisture due to evaporation, to roughly zero below the cloud. In the cloud top region it exceeds 100 $\mathrm{W\,m^{-2}}$ (subject to very large variability). It is not clear what are the contributions of radiative and evaporative cooling towards the observed heat fluxes there. LEG3 was performed close to the cloud top but neither exactly at the interface nor inside the EIL. Although CTEI parameter $\kappa$





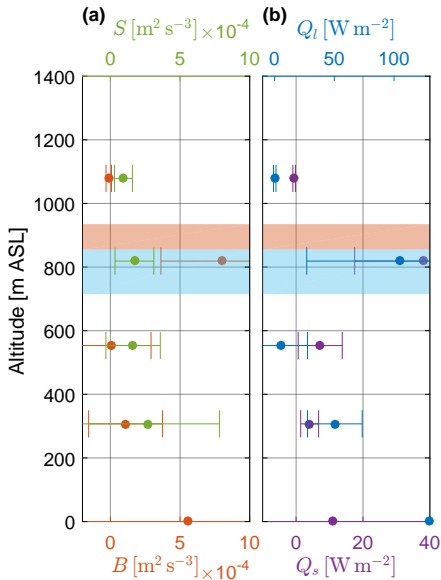

**Figure 11.** (a) TKE production by buoyancy $B$ and shear $S$, (b) sensible $Q_s$ and latent $Q_l$ heat fluxes in the coupled STBL (flight #5). The lowest dot denotes the parameterized surface value obtained with COARE 3.0 algorithm.

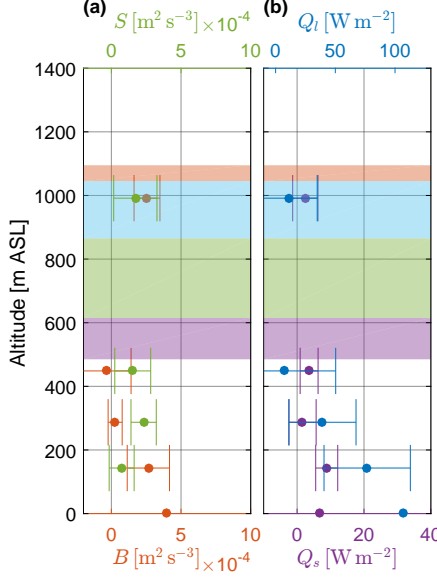

**Figure 12.** As in Fig. 11 but in the decoupled STBL (flight #14).

significantly exceeds the critical value (see sec. 3.2), which suggests the importance of evaporation, radiative cooling might still be dominant as in the study of Gerber et al. (2016).





In the decoupled STBL observed in flight #14 (Fig. 12), production terms are of the same order as in the coupled case. The COARE algorithm provides $B = 4.0 \cdot 10^{-4}$ m$^2$ s$^{-3}$, $Q_s = 6.7$ W m$^{-2}$, $Q_l = 107$ W m$^{-2}$ at the surface. $B$ decreases with height turning into weak buoyancy consumption at the transition. This can be considered an important signature of decoupling.

Above, in the cloud, $B$ is again positive, yet significantly smaller ($2.6 \cdot 10^{-4}$ m$^2$ s$^{-3}$) than at similar location in the coupled STBL. Shear production is present in the SML and at the transition as well as in the cloud top region.

Sensible heat flux in the decoupled boundary layer is relatively small reaching maximum of $\sim$10 W m$^{-2}$ at $\sim$140 m. Latent heat flux features near linear decrease with height from the maximum of $\sim$100 W m$^{-2}$ at the surface to roughly zero at the transition. Both sensible and latent heat fluxes observed in the cloud (LEG5) are small, in contrast to the coupled case. One may

speculate that the drivers of convection, i.e. radiative and evaporative cooling, are not efficient in this situation which might have been one of the reasons why decoupling occurred. This sounds consistent with rather moderate $B$ in the cloud. Another observation is that the moisture delivery from the ocean surface to the cloud might be more difficult in the decoupled STBL as $Q_l$ vanishes at much lower height in relation to the cloud base than in the coupled case.

## 5.3 TKE dissipation rate

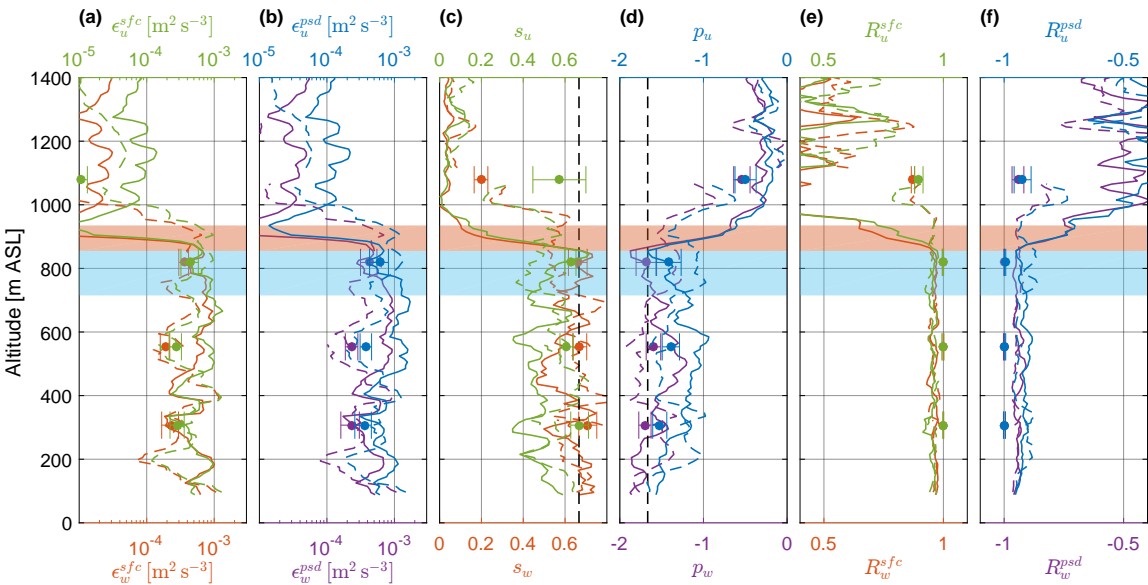

**Figure 13.** TKE dissipation rate and inertial range scaling in the coupled STBL (flight #5): (a), (b) dissipation rate $\epsilon$, (c), (d) fitted exponents $s$ and $p$, (e), (f) correlation coefficient $R$. Superscripts $sfc$ and $psd$ denote the structure function and power spectrum methods, respectively. Subscripts $u$ and $w$ denote horizontal and vertical velocity components, respectively. Dissipation rates for LEG2 which are not visible in panels (a), (b) are smaller than $10^{-5}$ m$^2$ s$^{-3}$.

Measurements in the coupled STBL during flight #5 (Fig. 13) indicate relatively small variability of TKE dissipation rate throughout the boundary layer depth and substantial decrease right above the cloud top. The values fluctuate by roughly



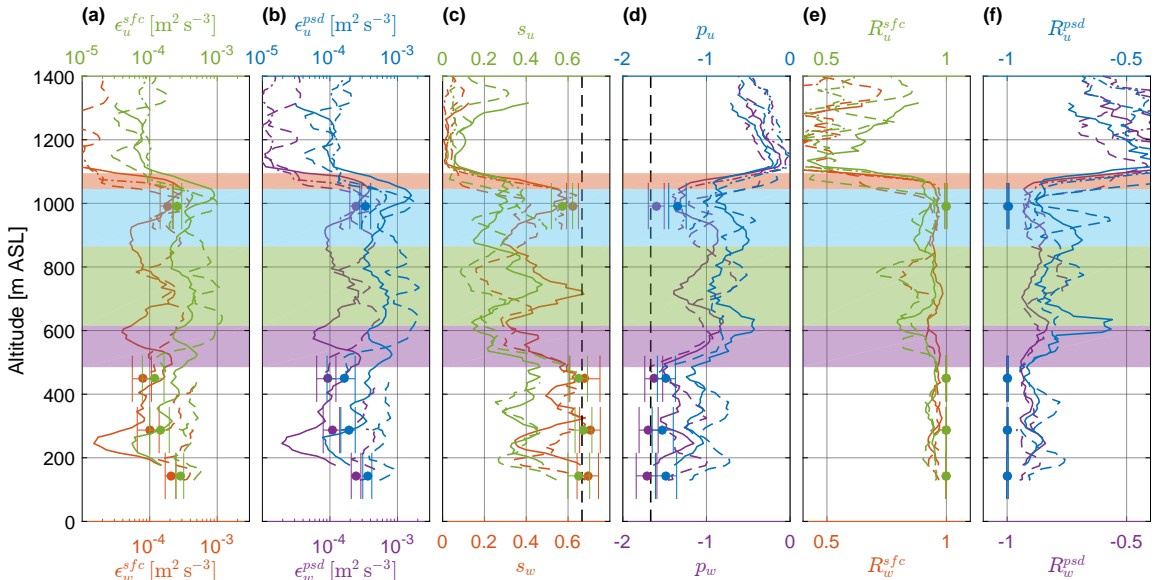

**Figure 14.** As in Fig. 13 but for the decoupled STBL (flight #14).

1 order of magnitude, between $10^{-4}$ m² s⁻³ and $10^{-3}$ m² s⁻³. Importantly, those variations do not correlate between the PROFs, hence they are the manifestation of some intermittency and random effects involved in airborne sampling rather than any systematic stratification. Among the LEGs, the highest dissipation rate was observed in the one close to the cloud top, where also substantial buoyant production of TKE was revealed (see sec. 5.2). On the other hand, continuous profiles of $\epsilon$ derived from PROFs do not show significant difference between the cloud and the subcloud part. It suggests that even though the TKE might be produced at specific places it is probably redistributed well by the circulation across the STBL before being dissipated by viscosity (c.f. transport analysis by Kopec et al. (2016)).

Inside the STBL, the exponents of structure function $s$ (sec. 4.3.1) and of power spectra $p$ (sec. 4.3.2) are close to their theoretical values (2/3 and $-5/3$, respectively), in striking contrast to the FT. Individual deviations occasionally reach 40 % in the STBL. On average, the deviations are a bit smaller inside the SCL than in the SBL (see Table A1). Typically, SFCs and PSDs seem to be flatter than the theory predicts (absolute values of $s$ and $p$ smaller than theoretical). Such behavior might be attributed to the non-homogeneity and non-stationarity of turbulence and different stages of its development, e.g. decay (Vassilicos, 2015). When different velocity components are concerned, SFCs and PSDs of vertical fluctuations follow Kolmogorov theory closer than the longitudinal, signaling some anisotropy in turbulence energy cascade.

Correlation coefficients $R^{sfc}$ and $R^{psd}$ (sec. 4.3) are close to unity in the coupled STBL. This implies both the SFC and the PSD can be considered linear in log-log coordinates in the assumed inertial range of scales. The correlation is higher for LEGs than for PROFs due to better averaging. It sharply decreases across the EIL, suggesting that in the FT the assumptions involved in the derivation of $\epsilon$ are not satisfied. Therefore, $\epsilon$ estimates above the boundary layer cannot be considered credible





(Akinlabi et al., 2019). On the other hand, inside the STBL the observed forms of SFC and PSD are reasonably consistent with theoretical predictions.

Measurements in the decoupled STBL during flight #14 (Fig. 14) present lower values of $\epsilon$ and more variability with respect to height. PROF-derived results averaged across the sublayers increase from the SML up to the SCL (see Table A2). Such a trend is consistent for all derivation methods and velocity components, despite differences in the absolute values among them.

The LEG-derived $\epsilon$ decreases with height, from the surface up to the transition.

Vertical profiles of the fitted exponents $s$ and $p$ reveal internal layering of the STBL. In contrast to the coupled case, all PROF-derived exponents deviate significantly from theoretical values. The deviations are appreciably smaller in the SML than in the SBL and the SCL, clearly demonstrating that turbulence in the upper part of decoupled STBL is further from Kolmogorov's concepts than in the lower part. The parameters inside the SCL and the SBL are comparable, suggesting there is an efficient

circulation and mixing across them. Those facts were expected, taking into account our analysis of stratification (sec. 3.4) and TKE production (sec. 5.2). Most probably, turbulence generated in the cloud top region is redistributed by the large eddies and the transport terms of the TKE balance equation (Stull, 1988) across the SCL and the SBL. Though, the properties of such turbulence are remarkably far from the Kolmogorov theory assuming homogeneity, isotropy and stationarity. In the light of this observation, the dissipation rates obtained with the methods based on the theoretical inertial range scalings can become

questionable. The assumptions are better resembled by the conditions in the lowermost part of the atmosphere, albeit they are still distant from being exactly fulfilled. The profiles of $R^{sfc}$ and $R^{psd}$ are in agreement with the above hypothesis suggesting different character and origin of turbulence in the upper and lower part of the STBL. The absolute values are smaller than in the coupled case. In the SBL and the SCL the correlation is even quite poor at some particular heights. In contrast to the PROFs, the LEG-derived exponents stay mostly close to 2/3 or -5/3, accordingly, while the correlations are close to one. Unfortunately,

none of the horizontal segments was performed in the SBL.

### 5.4 Anisotropy

The coupled STBL sampled in flight #5 features bulk anisotropy ratios predominantly in the range between 0.5 and 1.0 (Fig. 15). The variance anisotropy is largest (0.9) for the horizontal segment inside the cloud, close to its top where the turbulence is efficiently generated by buoyancy (sec. 5.2). In the SBL the values are a bit smaller. Despite substantial local

fluctuations observed in $A_{\epsilon}^{sfc}$ and $A_{\epsilon}^{psd}$, their average level can be considered constant across the boundary layer. There is very little difference between the SBL and the SCL. The SFC-derived anisotropy ratio is relatively close to unity, suggesting near isotropic conditions. However, the PSD-derived ratio, typically around 0.6, seems to indicate the dominant role of horizontal fluctuations. The reason for such a discrepancy between the methods is not clear. It can be related to the bias in the estimation of dissipation rates between them (c.f. Wacławczyk et al., 2020). Nevertheless, both anisotropy measures indicate no internal

layering inside the STBL. In the FT, under static stability and weak turbulence production, horizontal motions dominate.

In the decoupled STBL investigated in flight #14, bulk anisotropy ratios are on average smaller than in the previous case (Fig. 16), signaling prevalence of horizontal fluctuations over vertical ones. $A_2^{var}$ is the largest in the surface layer (reaching 0.72), smaller in the cloud (0.54) and close to the transition (0.41) between the two circulation systems, cloud-driven and





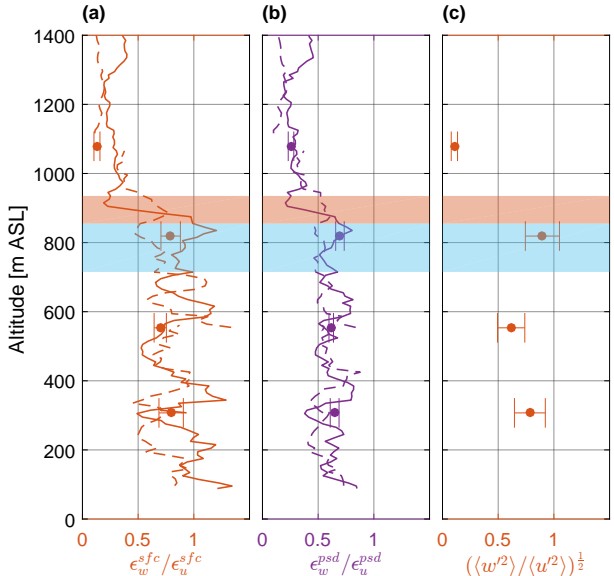

**Figure 15.** Anisotropy ratios in the coupled STBL (flight #5).

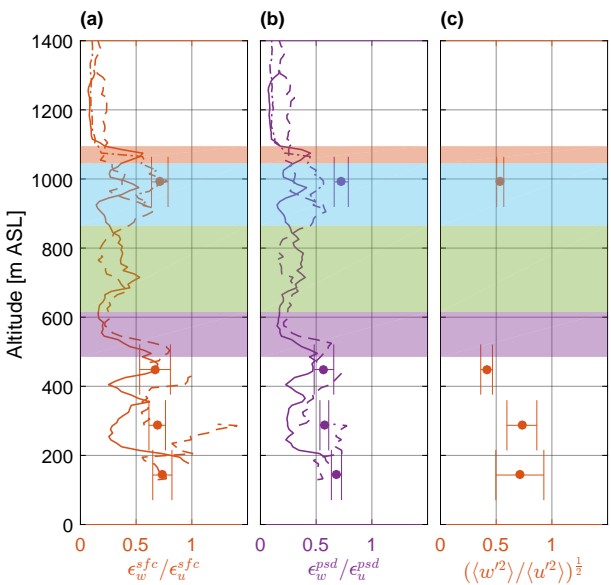

**Figure 16.** Anisotropy ratios in the decoupled STBL (flight #14).

surface-driven. Dissipation-derived anisotropy ratios imply the separation of the STBL into two parts with the border in the
TSL. In the upper part, covering the SCL and the SBL, $A_\epsilon^{sfc}$ and $A_\epsilon^{psd}$ are visibly smaller than in the SML. Again, the PSD-





derived rate is systematically lower than the SFC-derived, but the discrepancy is not as pronounced as in the case of flight #5. Importantly, the change at ∼500 m correlates well with the change in the fitted SFC and PSD exponents (see sec. 5.3) as well as with the gradient of specific humidity (see sec. 3.4). This fact confirms the hypothesis involving two major circulation circuits dividing the STBL into two parts which are internally relatively well-mixed but feature turbulence of different character. In the

SML, turbulence seems to be more vigorous and isotropic than in the SCL and the SBL.

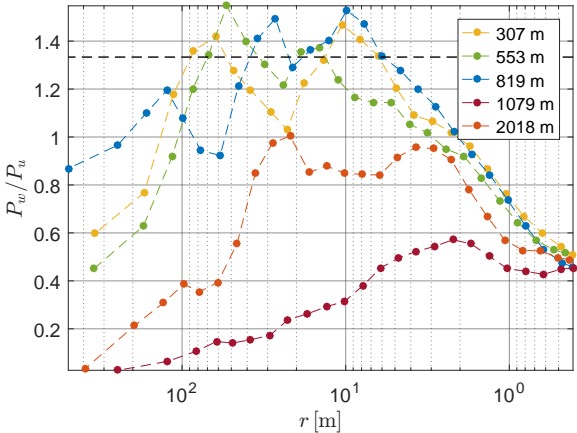

**Figure 17.** Spectral anisotropy ratio in the coupled STBL (flight #5). LEG-derived curves are labeled according to the altitude. The horizontal dotted line denotes the 4/3 level expected for isotropy in inertial range.

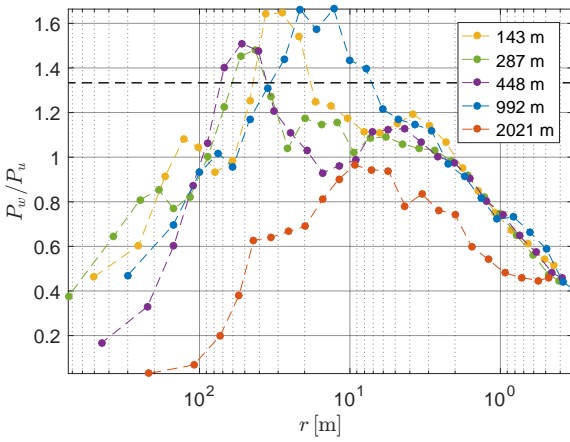

**Figure 18.** As in Fig. 17 but for the decoupled STBL (flight #14).

Spectral anisotropy ratios in the coupled STBL presented in Fig. 17 are of similar form for all three LEGs inside the boundary layer, contrasting with those performed in the FT. Inside the STBL, $A_P$ matches approximately the theoretical value of 4/3 in





the range of 5-100 m, indicating isotropic properties of turbulence in the inertial subrange of the energy cascade. The anisotropy ratios gradually decreases for larger scales which are of the order of the integral lengthscale (see sec. 5.5). Wavelengths of the size of a few hundred meters, which is close to the boundary layer depth (about 850 m), might be additionally influenced by the proximity of the bottom and top interfaces limiting their vertical extent. On the opposite side of the spectrum (short wavelengths), $A_P$ can be affected by the differences in the spectral transfer functions of the sonic anemometer for different velocity components (Kaimal et al., 1968). Similar effect was briefly described by Siebert et al. (2006b). In the FT, $A_P$ hardly reaches 1.0 because vertical excursions are damped by stability. In case of LEG2, it is particularly small, probably because that level was very close to the immensely stable temperature inversion.

In the decoupled STBL sampled in flight #14 (Fig. 18), $A_P$ follows similar pattern as observed in flight #5. Nonetheless, maximum values are higher, reaching up to 1.7 at the scale of 20-40 m in LEG1 and LEG5 which are the lowest and highest segment inside STBL. One may speculate those scales, featuring prevalence of vertical fluctuations, are related to the typical size of surface layer plumes and to the typical size of cloud top downdrafts manifested as cloud holes (Gerber et al., 2005). The range of scales where $A_P$ indicates conditions close to local isotropy is narrower than in the coupled STBL. On the side of large scales, $A_P$ falls below the theoretical 4/3 already at around 70 m for the two central LEGs and at around 50 m for the two peripheral LEGs (regarding the perspective of the STBL). This observation can be related to the integral lengthscales which are smaller than in flight #5 for the most part (see sec. 5.5). What is more, the depths of the two sections of the boundary layer corresponding to the supposed circulation circuits (~500 m) are also smaller than the total depth of the coupled STBL (~850 m).

## 5.5 Lengthscales

In the coupled STBL, the estimated integral scales vary around 100-150 m (Fig. 19). The longitudinal scale $L_u$ increases, whereas the vertical $L_w$ decreases with height. The ratio $L_w/L_u$ decreases from about 1.3 in the lowest LEG to about 0.5 (as expected for isotropic turbulence) close to the cloud top. The variability of integral scales among the subsegments of the LEGs is extensive, reflecting poor averaging on relatively short distances which prevents accurate calculation of decorrelation length.

Estimated Taylor microscales fit into the range of 30-80 cm and decline with height from the middle to the top of the STBL. As predicted, the longitudinal $\lambda_u$ are larger than the vertical $\lambda_w$. Their ratio $\lambda_u/\lambda_u$ equals $\sqrt{2}$ (corresponding to isotropy of small-scale turbulence) only in the cloud LEG and is larger below. We may speculate that the turbulence is close to isotropic at the time and location of generation but such isotropy might be broken in the process of transport. Kolmogorov microscale is almost constant across the STBL (~2 mm) which can be expected as it depends practically only on the dissipation rate (the viscosity changes only by a minor part in the lower atmosphere). There is also no major difference between the horizontal and vertical direction.

In the decoupled STBL, integral scales are significantly smaller in comparison to the previous case, hardly exceeding 100 m (Fig. 20). The longitudinal $L_u$ dominates over the vertical $L_w$, probably due to the separation of the circulation into two circuits and weak static stability which both limit the vertical extent of eddies and promote horizontal elongation. In contrast to the





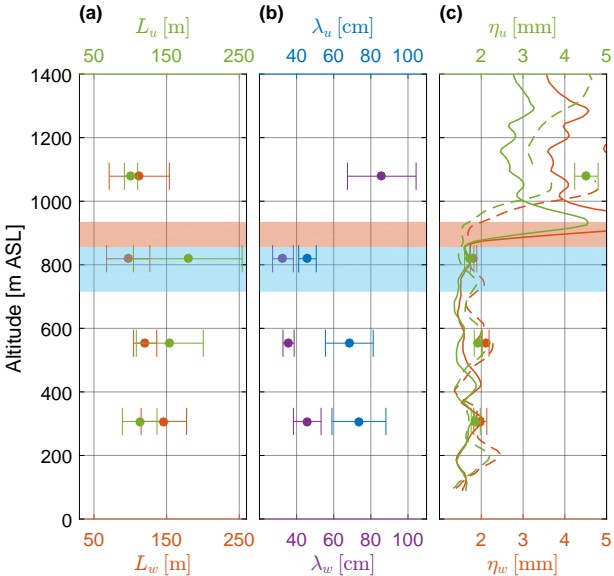

**Figure 19.** Turbulent lengthscales in the coupled STBL (flight #5): (a) integral scale $L$, (b) Taylor microscale $\lambda$, (c) Kolmogorov scale $\eta$. Subscripts $u$ and $w$ denote horizontal and vertical velocity components, respectively. Some of the results for LEG2 in the FT are out of the range presented.

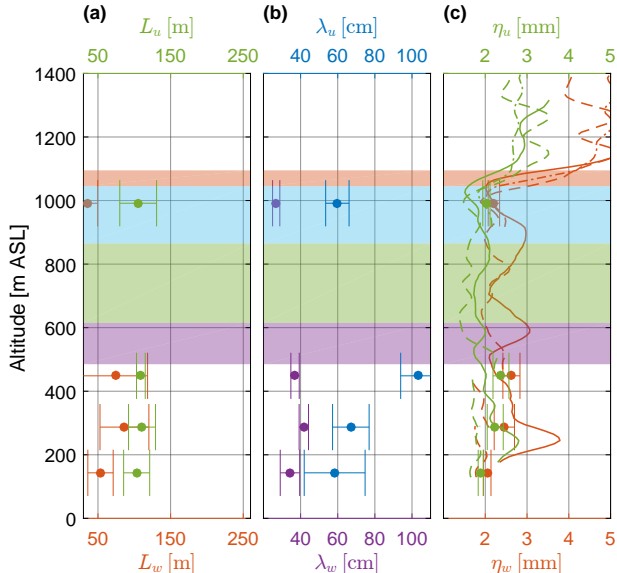

**Figure 20.** As in Fig. 19 but for the decoupled STBL (flight #14).





coupled case, the ratio $L_w/L_u$ equals about one half in the lowest LEG close to the surface which is, however, again the location of intensive TKE production.

Taylor microscale is mostly of the same order as in the former case. In the transition zone and in the cloud, substantial detachment between the longitudinal and the vertical can be observed. $\lambda_u/\lambda_u$ is significantly larger than expected for isotropic
turbulence. This effect is most pronounced in LEG3 close to the transition. We may speculate it might be the consequence of decaying turbulence – far from the production in the cloud and at the surface, the TKE is here dissipated and consumed by weak buoyant stability (sec. 5.2). Kolmogorov scale visibly fluctuates but on average stays close to constant across the STBL. In contrast to the coupled case, there is some difference between $\eta_u$ and $\eta_w$ which directly relates to $A_\epsilon^{sfc}$ discussed in sec. 5.4.

## 6   Summary and discussion

Two cases of marine stratocumulus-topped boundary layer, coupled (CP) and decoupled (DCP), have been compared in terms of stratification and turbulence properties. The observations were performed in summer in the region of Eastern North Atlantic with the use of the helicopter-borne platform ACTOS. Its moderate true air speed in combination with closely collocated fast-response instruments provides high spatial resolution measurements of turbulent fluctuations of wind velocity, temperature and humidity. Similarities and differences between the two cases can be summarized as follows.

1. *Stratification*

   **CP** Conserved variables, $\theta_l$ and $q_t$, feature nearly constant profiles up to the capping inversion at $\sim$850 m. LCL can be considered consistent with cloud base height.

   **DCP** Above the relatively well-mixed SML, $\theta_l$ slowly increases with height up to the capping inversion at $\sim$1050 m, indicating weak stability. There is a significant gradient of $q_t$ in the TSL. LCL is close to the observed CB in the
SBL only. Decoupling of the STBL was detected according to simple thermodynamic criteria.

   In both cases winds are moderate and appreciable wind shear is observed across the cloud top and the EIL.

   2. *TKE production*

   **CP** TKE is efficiently generated by buoyancy with simultaneous importance of in-cloud and surface processes. Buoyancy production follows typical STBL profile: decreases with height from the surface upwards, vanishes or turns
slightly negative below cloud base, to be again substantial inside the cloud due to latent heat release and diabatic cooling.

   **DCP** TKE is generated by buoyancy at the surface and $B$ decreases with height to zero at the SML top, turning into buoyancy consumption in the TSL. In the cloud $B$ is weaker than at the surface, about three times smaller than for the CP. Buoyancy effects can be also deduced from spectral anisotropy in the uppermost and lowermost boundary
layer LEGs which suggests dominance of vertical motions in scales of 10-40 m.





The contribution of shear to TKE production is not negligible in both cases. This result can be partly artifact because only the longitudinal term could be evaluated and due to inaccurate estimation of horizontal wind gradient involved in shear term.

3. *Heat fluxes*

In both cases latent heat flux qualitatively resembles the profile of $B$ which suggests important contribution of moisture to buoyancy. $Q_l$ is large at the ocean surface and decreases to zero at similar level as the minimum of $B$. Sensible heat flux is positive throughout observed layers but mostly smaller than $Q_l$.

**CP** $Q_l$ and $Q_s$ are positive and of significant magnitude close to the cloud top which can be attributed to diabatic cooling (radiative and/or evaporative).

**DCP** $Q_l$ and $Q_s$ are small close to the cloud top, about an order of magnitude weaker than for the CP. Additionally, $Q_l$ vanishes at the level much lower in relation to the cloud base which might disturb moisture delivery from the ocean to the SC.

4. *Turbulent fluctuations*

In both cases TKE is dominated by the contribution of horizontal velocity fluctuations. Variances of temperature and
humidity are significant in the regions where mixing between air volumes of different properties occurs – due to local gradients or sources/sinks, i.e. at the cloud top, at the surface and at the transition in the DCP.

**CP** Maximum TKE is found in the middle of the SBL which together with positive $\langle w'^3 \rangle$ at this level point out the role of surface-related factors in generating convection. Vertical velocity variance suggests the profile somewhat different than the convective similarity scaling. In cloud, $\langle u'^2 \rangle$ and $\langle w'^2 \rangle$ are almost equal implying isotropic
conditions.

**DCP** The SML follows the structure of a typical mixed layer with shear (c.f Stull, 1988). Updrafts are stronger than downdrafts. TKE, $\langle T'^2 \rangle$ and $\langle q_v'^2 \rangle$ are largest close to the transition. In cloud, fluctuations are relatively weak, in particular $\langle w'^2 \rangle$, in concordance with limited $B$ and small heat fluxes.

5. *TKE dissipation*

**CP** Derived $\epsilon$ varies weakly throughout the height, i.e. despite accidental variations no systematic layering can be observed. Although TKE is efficiently produced by buoyancy in the cloud and at the surface, it is probably redistributed well across the depth before being dissipated by viscosity. The form of SFCs and PSDs is reasonably consistent with theoretical predictions for inertial range scaling in homogeneous, isotropic, stationary turbulence (Kolmogorov, 1941). Though, less steep scaling (smaller absolute values of $s$ and $p$) can be found at some places
in the SBL.





**DCP** Derived $\epsilon$ is smaller than in the CP and features differences between the sublayers. Despite relatively high $B$ at the surface, similar to the CP, average $\epsilon$ in the SML is smaller than in the SCL. Importantly, SFCs and PSDs scaling in inertial range considerably deviates from the theoretical. Such behavior is characteristic for decaying turbulence (less energy than expected in large scales). These deviations are more pronounced and more variable in the SCL and SBL in comparison with the SML, underlining different character of turbulence in the upper and lower part of the DCP. Probably, TKE generated in the surface region and in the cloud, respectively, is redistributed in the two circulation zones separately, without major transport through the transition.

Discrepancies between PROF-derived and LEG-derived quantities result from the contrast between local and mean turbulence characteristics. The observed relative tendencies are consistent among derivation methods and velocity components, in spite of discrepancies in the absolute values.

6. *Anisotropy of turbulence*

   **CP** Derived anisotropy ratios indicate that turbulence is relatively close to isotropy. This condition is met best in the cloud where significant TKE production occurs.

   **DCP** The degree of anisotropy varies between the sublayers. In the uppermost part (SCL and SBL) horizontal small-scale velocity fluctuations dominate over the vertical. This effect is less pronounced in the SML. The change in anisotropy ratios in the TSL coincides with the difference in $s$ and $p$ right below the strong $q_v$ gradient.

7. *Lengthscales of turbulence*

   Integral lengthscales of the order of $100\,\mathrm{m}$ show that turbulent eddies are substantially smaller than the depths of STBL or decoupled sublayers. Thus, they can be considered small enough to be transported by larger circulations.

   **CP** In the middle SBL, $w'$ is correlated on longer distances than $u'$, while the opposite holds in the SCL. This agrees with the supposed form of circulation in the boundary layer, i.e. downdrafts originated at cloud top and updrafts originated at the surface pair in the middle and diverge horizontally in the vicinity of top and bottom boundaries.

   **DCP** Integral lengthscales are smaller than in the CP. In accordance with anisotropy ratios, $L_u$ is larger than $L_w$. The same holds for Taylor microscales. The difference between $\lambda_u$ and $\lambda_w$ is particularly pronounced close to the transition. It seems that even smaller turbulent eddies there are elongated in horizontal.

   Interestingly, $L_w/L_u \approx \frac{1}{2}$ implied by isotropy assumption holds only in the regions of intensive buoyant TKE production: in the cloud for the CP and close to the surface for the DCP. Kolmogorov scale is $\sim2\,\mathrm{mm}$ in both cases.

Most of our results concerning the coupled case are consistent with previous studies of SC dynamics (e.g. Nicholls and Turton, 1986; Duynkerke et al., 1995; Stevens et al., 2005; Kopec et al., 2016; Dodson and Small Griswold, 2021). In particular, the $B$ profiles show that convection is driven both by cloud top cooling and by surface thermal instability. However, our results suggest the profile of $\langle w'^2 \rangle$ being somewhat different than the convective similarity scaling (Lenschow et al., 1980) but rather





having maximum in the cloud and minimum below it, with $A_2^{var}$ following the same behavior, similarly to Dodson and Small Griswold (2021). Together with high TKE and positive $\langle w'^3 \rangle$ in the middle SBL, this highlights the importance of surface process. It might be related to small cloud depth (relative to STBL depth) and net cooling at cloud top reduced during daytime

in comparison to often considered nocturnal SC. In contrast to the works listed above, we do not clearly observe the maximum of $\epsilon$ at the top and at the bottom of the STBL, but it is rather because others applied considerable horizontal averaging in comparison to local variability captured in our PROFs.

Our observations in the decoupled STBL summarized in points 1-4 fit well into the range of conditions reported in the literature, in particular the properties of the SML. Buoyant TKE production is positive in the cloud, while there is a region of

negative $B$ around the transition (Nicholls, 1984; Nicholls and Turton, 1986; Turton and Nicholls, 1987; Durand and Bourcy, 2001). Moreover, $Q_l$ decreases from the surface to zero at the transition and it is substantially larger than $Q_s$ in the SML (Nicholls, 1984; Tjernström and Rogers, 1996; De Roode and Duynkerke, 1997; Lambert and Durand, 1999; Durand and Bourcy, 2001). However, Lambert and Durand (1999) dispute the nearly linear character of this decrease, suggesting rather sharp gradient right at the SML top. Comparable to Nicholls (1984), our variances $\langle T'^2 \rangle$, $\langle q_v'^2 \rangle$ are significant close to the

surface and have local minimum in the middle SML where in turn $\langle w'^2 \rangle$ is relatively large. As in De Roode and Duynkerke (1997), $\langle w'^3 \rangle$ is positive in the SML and nearly zero in the SCL, although the LEGs were rather too short to ensure statistical significance of those results. On the other hand, we did not collect enough data in the SCL and SBL to judge whether they together exhibit upside-down convective scaling as in Nicholls and Turton (1986); Tjernström and Rogers (1996); De Roode and Duynkerke (1997).

The results of our comparison between coupled and decoupled STBL are in agreement with the common concept of the dominant mixing patterns in such boundary layers (e.g. Wood, 2012). Decoupling occurs when the thermally driven circulation weakens to the level that it cannot mix air throughout entire depth. Then, STBL separates into two parts: cloud driven and surface driven. Explaining the particular mechanism of decoupling operating in our case is beyond the scope of this study and would require more complete data on airmass history. Nevertheless, "deepening-warming" mechanism (Bretherton and Wyant,

1997) seems plausible. Such conclusion was reached by Kazemirad and Miller (2020) who modeled lagrangian evolution of STBL on synoptic scale in the period including our measurements. Deepening-warming is typical for the ENA region where air masses are advected over progressively warmer waters. The most important driver for this process is the increasing ratio of surface latent heat flux to net radiative cooling in the cloud. The former was indeed relatively large, the latter was probably reduced by daytime solar heating. In addition, some precipitation was reported shortly before the flight and evaporative cooling

could have contributed to stabilizing the lower STBL. Finally, decoupling occurs more readily for large entrainment efficiency. Derived $B$ is weak in the cloud, much smaller than in the coupled one, which might be the result of enhanced entrainment warming offsetting radiative cooling (c.f. De Roode and Duynkerke, 1997).

The important novelty of our work are the results on small-scale turbulence (points 5-7 of the summary). As far as we know, local $\epsilon$ profile, inertial range scaling exponents and anisotropy ratios were not addressed in the context of STBL coupling

before. Based on the observations, we hypothesize that turbulence is redistributed across the depth of the CP but in case of





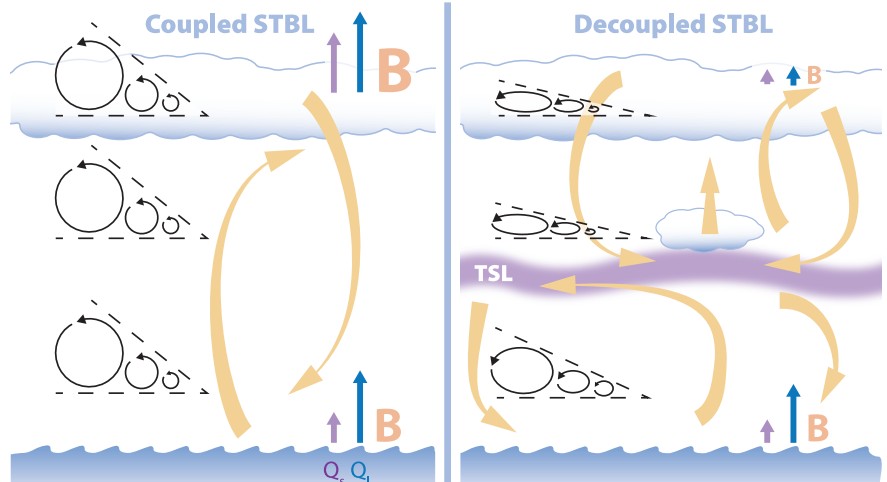

**Figure 21.** Schematic of main processes in the coupled (left) and decoupled (right) STBL: primary circulation (yellow arrows), turbulence eddy cascade (circular arrows confined in an angle with extent proportional to inertial range scaling exponent $p$), TKE buoyancy production (red B letter of size proportional to strength), sensible and latent heat fluxes (purple and blue arrows, respectively, of length proportional to strength) at the surface and in the cloud top region.

the DCP primarily in the sublayer where it was generated. Therefore, specific microscopic properties – TKE dissipation rate, inertial range scaling and anisotropy – can differ between the parts of the DCP.

We consider it important to emphasize often omitted distinction between *circulation* and *turbulence*. By *circulation* we understand motions responsible for mixing across relatively deep layers, of vertical scales comparable to PBL depth. They usually originate from thermally driven plumes, sinking from cloud top or rising from the surface. Circulation might take form of organized structures of downdrafts and updrafts (resembling Rayleigh-Bennard convection cells). Those correspond to the peak in vertical velocity spectra, typically at $\sim$1 km in STBL (Lambert et al., 1999). *Turbulence* features cascade of eddies with universal scaling properties (Kolmogorov, 1941), spanning from the integral lengthscale ($\sim$100 m in STBL) down to the Kolmogorov scale ($\sim$1 mm) where TKE is dissipated by viscosity. Such turbulence can be generated by flow instabilities at specific locations (here typically close to the surface and cloud top) and distributed by circulation within STBL, alongside other constituents. Importantly, the variances and fluxes estimated in our study include contributions of both phenomena. Circulation is only partly resolved as we applied the cutoff of $\sim$1 km in Reynolds decomposition due to limited length of LEGs. Similar issue was also raised by De Roode and Duynkerke (1997). The advantage of our work is a good representation of turbulence because we resolve significant portion of the inertial range. Main processes operating in the coupled and decoupled STBLs, including circulation and turbulence, are schematically illustrated in Fig. 21.

Both turbulence and circulation can contribute to vertical transport of heat and moisture which is crucial for maintaining SC. In the decoupled STBL, transport by turbulence through the transition is rather limited. However, we speculate it can be efficiently realized by a small number of updrafts which are strong and moist enough to penetrate the conditionally unstable





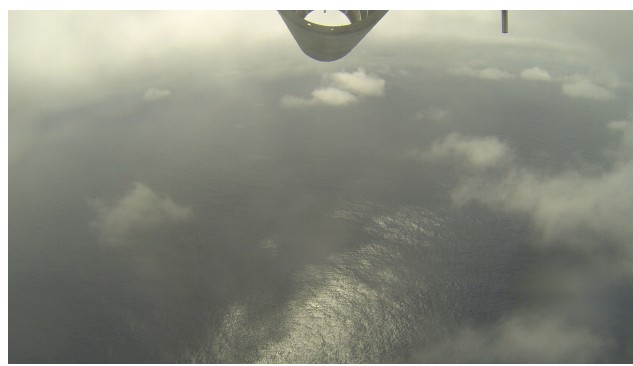

**Figure 22.** Cumulus clouds under stratocumulus in the decoupled STBL. Photograph was taken during PROF5 of flight #14 by the camera mounted on the bottom of ACTOS.

TSL (measured $\Gamma_T$ = -7.1 K km$^{-1}$, moist adiabatic $\Gamma_T$ = -4.7 K km$^{-1}$), reach their LCL and form cumulus clouds. The image

of those cumuli was captured by a camera onboard ACTOS (Fig. 22). Based on the series of images from PROF5, we estimated the cloud base height $\sim$660 m (equal to mean LCL in the TSL) and cloud depth $\sim$100 m. None of those cumuli was penetrated by ACTOS, so it is not possible to distinguish dynamic effects responsible for their formation. Detailed analysis of vertical transport calls for high-resolution numerical simulations to be setup with the help of our results.

The onset of cumulus convection depends on the properties of the TSL which is then imortant for overall STBL dynamics.

However, it is a challenge to conduct relevant systematical climatological analysis of TSL existence and proprties due to limited number of observations. The reason is often insufficient resolution of routine radiosoundings. For instance, the layer of the strongest gradient in $q_v$ (550-600 m) penetrated in PROF5 features the differences of $\Delta\theta$ = 0.4 K, $\Delta q_v$ = 1 g kg$^{-1}$ (equivalent to $\Delta RH$ = 8 %) and $\Delta LCL$ = 160 m. With the ascent rate of $\sim$5 m s$^{-1}$ and sampling interval of $\sim$2 s, a hygrometer with the time constant of a second and the accuracy of a single percent in RH would be desired. Moreover, TSL is not exactly flat but

rather undulated as suggested by our data of LEG3. Therefore, even aircraft measurements may fail to properly capture local conditions. This was pointed out already by (Turton and Nicholls, 1987, p. 997) who underlined the role of good observation strategy: "While cloud layer decoupling is predicted to occur quite often, the consequential modification of the horizontally averaged vertical thermodynamic structure remains fairly small. (...) Data averaged in this way will appear 'nearly well-mixed' whether separation has occurred or not. A more detailed analysis of individual profiles and turbulence data is necessary to

determine whether decoupling has taken place".



*Data availability.* The whole dataset collected within the ACORES field project is planned to be archived on the PANGEA server for public access. The data used in the present study is also available from the authors upon request.

*Author contributions.* All the authors participated in the instrument preparation, measurements and data postprocessing within the ACORES project where H.S. was the principal investigator. J.L.N. designed the presented analysis with advice from S.P.M and H.S. The analysis was performed by J.L.N. with contributions from K.E.S. and with guidance of H.S. and S.P.M. J.L.N. wrote the manuscript with contributions from S.P.M. and H.S.

*Competing interests.* The authors declare that they have no conflict of interest.

*Acknowledgements.* The field campaign ACORES was supported by several grants of the Deutsche Forschungsgesellschaft (DFG, with grants SI 1543/4-1, WE 1900/33-1, WE 2757/2-1, and HE 6770/2-1) and Polish National Science Centre (grant agreement 2013/08/A/ST10/00291). J.L.N. acknowledges the one year grant awarded by the German Academic Exchange Service (DAAD) for his research visit to Leibniz Institute for Tropospheric Research. We acknowledge the use of imagery from the Worldview Snapshots application (https://wvs.earthdata.nasa.gov), part of the Earth Observing System Data and Information System (EOSDIS). The authors are also grateful to Dr. Marta Waclawczyk for discussions on the manuscript and to Katarzyna Nurowska for drawing the sketch in Fig. 21.

## Appendix A: Average conditions in the sublayers

Average meteorological parameters and turbulence properties inside the sublayers of the atmosphere are summarized in Tables A1 and A2 for the coupled (flight #5) and decoupled (flight #14) case, respectively. The selection of the sublayers is explained in sec. 3. The average values were obtained from the data of the same PROF which served for sublayer selection, i.e. PROF1 in coupled case, PROF5 in decoupled case. $\Gamma_T$, $\mathrm{Nb}^2$ and $\mathrm{Sh}^2$ were calculated by estimating derivatives over sublayer depth. Other parameters were simply averaged in the relevant altitude range.





**Table A1.** Average conditions inside the sublayers in the case of coupled STBL (flight #5).

| Parameter | SBL | SCL | EIL | FTL |
|---|---|---|---|---|
| Height [m] | 0 - 715 | 715 - 855 | 855 - 935 | 1005 - 1385 |
| $T$ [$^\circ$C] | 16.24 | 12.59 | 14.53 | 14.41 |
| $\theta_l$ [$^\circ$C] | 17.62 | 17.52 | 20.59 | 23.54 |
| $\Gamma_T$ [K km$^{-1}$] | -10.9 | -10.1 | 73.9 | -7.2 |
| $q_t$ [g kg$^{-1}$] | 9.53 | 9.43 | 3.19 | 3.89 |
| $U$ [m s$^{-1}$] | 5.3 | 5.0 | 6.5 | 6.8 |
| $dd$ [$^\circ$] | 337 | 330 | 329 | 323 |
| LCL [m] | 814 | 845 | 3363 | 3130 |
| Nb$^2$ [$10^{-4}$s$^{-2}$] | -0.4 | -0.6 | 15.4 | 0.7 |
| Sh$^2$ [$10^{-4}$s$^{-2}$] | 0.0 | 0.3 | 5.1 | 1.0 |
| $\epsilon_w^{sfc}$ [$10^{-4}$m$^2$ s$^{-3}$] | 5.6 | 6.1 | 1.9 | 0.2 |
| $\epsilon_u^{sfc}$ [$10^{-4}$m$^2$ s$^{-3}$] | 6.5 | 6.6 | 2.2 | 0.8 |
| $\epsilon_w^{psd}$ [$10^{-4}$m$^2$ s$^{-3}$] | 5.6 | 5.1 | 1.5 | 0.3 |
| $\epsilon_u^{psd}$ [$10^{-4}$m$^2$ s$^{-3}$] | 9.2 | 8.5 | 2.6 | 1.2 |
| $s_w$ | 0.61 | 0.67 | 0.29 | 0.03 |
| $s_u$ | 0.47 | 0.55 | 0.34 | 0.05 |
| $p_w$ | -1.53 | -1.70 | -1.10 | -0.31 |
| $p_u$ | -1.25 | -1.42 | -1.03 | -0.23 |
| $R_w^{sfc}$ | 0.96 | 0.97 | 0.79 | 0.37 |
| $R_u^{sfc}$ | 0.95 | 0.96 | 0.87 | 0.42 |
| $R_w^{psd}$ | -0.94 | -0.95 | -0.81 | -0.49 |
| $R_u^{psd}$ | -0.91 | -0.93 | -0.81 | -0.41 |
| $\epsilon_w^{sfc}/\epsilon_u^{sfc}$ | 0.87 | 0.94 | 0.54 | 0.28 |
| $\epsilon_w^{psd}/\epsilon_u^{psd}$ | 0.62 | 0.63 | 0.41 | 0.29 |
| $\eta_w^{sfc}$ [mm] | 1.7 | 1.6 | 4.0 | 4.0 |
| $\eta_u^{sfc}$ [mm] | 1.6 | 1.6 | 3.0 | 2.9 |





**Table A2.** Average conditions inside the sublayers in the case of decoupled STBL (flight #14).

| Parameter | SML | TSL | SBL | SCL | EIL | FTL |
|---|---|---|---|---|---|---|
| Height [m] | 0 - 385 | 485 - 615 | 615 - 865 | 865 - 1045 | 1045 - 1095 | 1150 - 1400 |
| $T$ [°C] | 18.06 | 15.79 | 14.36 | 13.10 | 13.90 | 16.05 |
| $\theta_l$ [°C] | 18.81 | 19.20 | 19.61 | 20.16 | 22.37 | 26.29 |
| $\Gamma_T$ [K km$^{-1}$] | -10.1 | -7.1 | -7.4 | -2.9 | 84.6 | -5.2 |
| $q_t$ [g kg$^{-1}$] | 11.65 | 11.08 | 10.75 | 10.68 | 9.65 | 8.48 |
| $U$ [m s$^{-1}$] | 6.5 | 5.8 | 5.5 | 6.0 | 7.3 | 9.9 |
| $dd$ [°] | 314 | 308 | 314 | 322 | 322 | 325 |
| LCL [m] | 508 | 658 | 769 | 905 | 1328 | 2040 |
| Nb$^2$ [10$^{-4}$s$^{-2}$] | 0.5 | 0.9 | 0.1 | 1.5 | 28.0 | 1.5 |
| Sh$^2$ [10$^{-4}$s$^{-2}$] | 0.2 | 0.8 | 0.1 | 0.1 | 45.6 | 1.5 |
| $\epsilon_w^{sfc}$ [10$^{-4}$m$^2$ s$^{-3}$] | 0.6 | 1.1 | 1.2 | 1.3 | 1.8 | 0.1 |
| $\epsilon_u^{sfc}$ [10$^{-4}$m$^2$ s$^{-3}$] | 1.2 | 3.4 | 3.5 | 4.6 | 3.8 | 0.6 |
| $\epsilon_w^{psd}$ [10$^{-4}$m$^2$ s$^{-3}$] | 0.6 | 1.5 | 1.6 | 1.9 | 2.5 | 0.1 |
| $\epsilon_u^{psd}$ [10$^{-4}$m$^2$ s$^{-3}$] | 1.9 | 5.6 | 5.4 | 7.8 | 6.5 | 0.9 |
| $s_w$ | 0.52 | 0.45 | 0.47 | 0.44 | 0.38 | 0.04 |
| $s_u$ | 0.38 | 0.27 | 0.33 | 0.24 | 0.28 | 0.16 |
| $p_w$ | -1.39 | -1.10 | -1.10 | -1.11 | -1.05 | -0.34 |
| $p_u$ | -1.16 | -0.76 | -0.78 | -0.68 | -0.85 | -0.46 |
| $R_w^{sfc}$ | 0.95 | 0.96 | 0.96 | 0.96 | 0.87 | 0.30 |
| $R_u^{sfc}$ | 0.94 | 0.91 | 0.92 | 0.90 | 0.89 | 0.70 |
| $R_w^{psd}$ | -0.90 | -0.89 | -0.89 | -0.89 | -0.81 | -0.45 |
| $R_u^{psd}$ | -0.89 | -0.79 | -0.79 | -0.79 | -0.83 | -0.59 |
| $\epsilon_w^{sfc}/\epsilon_u^{sfc}$ | 0.48 | 0.31 | 0.34 | 0.29 | 0.48 | 0.09 |
| $\epsilon_w^{psd}/\epsilon_u^{psd}$ | 0.34 | 0.26 | 0.30 | 0.24 | 0.39 | 0.09 |
| $\eta_w^{sfc}$ [mm] | 3.0 | 2.6 | 2.5 | 2.5 | 2.4 | 5.6 |
| $\eta_u^{sfc}$ [mm] | 2.4 | 1.9 | 1.9 | 1.8 | 1.9 | 3.0 |



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
