# Peer review of "Coupled and decoupled stratocumulus-topped boundary layers: turbulence properties"

_Atmospheric Chemistry and Physics, 2021_

## Referee Comment (RC2)

**Referee comment on "Coupled and decoupled stratocumulus-topped boundary layers: turbulence properties" by Jakub L. Nowak, Holger Siebert, Kai-Erik Szodry, and Szymon P. Malinowski, Atmos. Chem. Phys. Discuss., https://doi.org/10.5194/acp-2021-214, 2021.**

**Anonymous Referee #2**

This paper presents high-resolution turbulent flux measurements obtained by the helicopter-borne platform ACTOS during the ACORES campaign. The focus of this study lies on the comparison of the stratification and turbulence properties in a (1) coupled and (2) decoupled marine stratocumulus-topped boundary layer. The paper is well written and the results are nicely related to previous work. I think that this study contributes to our understanding of turbulence in marine boundary layers. I recommend publication after addressing the comments below.

**General comments**

1. The paper provides interesting insights into the stratification and the turbulent properties of coupled (CP) and decoupled (DCP) marine stratocumulus-topped boundary layers. It would be interesting to have additional CP/DCP cases to investigate whether the same pattern is also observed in other cases. Are you planning to extend the analysis to more cases? It would be in particular interesting to have more data in the stratocumulus layer.

2. In Sect. 2.2, you introduce the different instruments that have been used in the study. I would like to see further discussion regarding the uncertainties of the measurements. Currently, uncertainties are not discussed and the error bars only include the variability of the data. This makes it difficult to assess the conditions inside the different sublayers in the CP and DCP case (e.g., add error bars in Table A1 and A2).

3. The paper contains many abbreviations (e.g., for the different sublayers). I understand that it makes sense to introduce these abbreviations, as they are frequently used in the paper. However, for the readers that are not familiar with these abbreviations, it can be hard to remember the definition of the different acronyms and to follow the text. Please check if all the abbreviations are necessary. For example, "ENA" and "CTEI" are only used 2-3 times and could be removed. Furthermore, I would suggest including the abbreviations of the different sublayers in the figures, in order to make it easier for the reader to identify them (see specific comments).

**Specific comments**

4. Page 5, caption Fig. 1 and caption Fig.3: Please add date and time of satellite image.

5. Page 5, Fig. 2 and Fig. 4: In Fig. 2 and Fig. 4 you show the time series of the ACTOS altitude. I think it would be beneficial to include more information regarding the cloud/BL structure. E.g., At what altitude is the cloud top/cloud base? You could indicated the different sublayers on the right side of the plot.

   Furthermore, you often refer to the different profiles (PROFs 1-5) and legs (LEGs 1-5) throughout the paper. You could consider adding the labels of the profiles and legs on top of the plot.

   In addition, the line style of the profiles is not evident in the figure due to the low contrast between the black line in the background and the black dotted line. I would suggest to remove the black line in the background or to change the color to get a better contrast.

6. Page 6, equation 1: You defined 'ql' as the liquid water content on page 4, line 103. The liquid water content is usually defined as mass of liquid water per volume of air (i.e. g m$^{-3}$). However, in equation 1 the liquid water mixing ratio (i.e. mass of liquid water per unit mass of air) should be used and not the liquid water content (see Betts 1973 or the following link: https://glossary.ametsoc.org/wiki/Liquid_water_potential_temperature). Please review your definition of 'ql' in the manuscript.

7. Page 7, line 150: According to J11, 'qt' should be the total water mixing ratio, which is defined by the sum of the liquid water mixing ratio and the water vapor mixing ratio (see also comment 6). Please review your definition of 'qt' in the manuscript.

8. Page 9, Fig. 5 and Fig. 6: As mentioned already in the general comment section, it is hard to remember the abbreviations of the different sublayers. In order to make it easier for the reader to follow and identify the different sublayers, I would suggest adding the abbreviations of the sublayers (color shaded areas) on the right side of the subplots (for all figures of this type; i.e. Fig. 5, 6, 9-16, 19-20). Furthermore, I would plot the lines on top of the shaded area to avoid any change in the line color (for example for LCL, qv).

9. Page 9, line 207: So are both the upper and the lower BL portion internally mixed? If yes, you could change the structure of the sentence as follows: "This suggests that both the upper and lower BL portion are internally mixed."

10. Page 14, line 113: You applied a moving window of 2 s to the profiles. How was the moving window of 2 s determined? Did you conduct sensitivity tests with different time windows?

11. Page 20, Fig.13: Why is there such a large discrepancy between some of the PROFs and LEGs properties (e.g., $s_u$, $R_u$, $R_w$) in the FTL?

12. Page 24, Fig. 17 and Fig. 18: I would use the same scale on the y-axis for Fig. 17 and Fig. 18 for better comparison between the coupled and decoupled case. Furthermore, I would suggest adding the sublayer in brackets next to the altitude.

13. Page 25, line 506: "Lengthscales" should be changed to "Length scales" throughout the paper.

14. Page 25, line 512 and line 524: One of the "$\lambda_u$" in the ratio should be replaced by "$\lambda_w$".

15. Page 32, line 669: Change "imortant" to "important"

16. Page 32, line 670: Change "proprties" to "properties"

---

## Author Comment (AC1)

**Authors' Response to the Anonymous Referee #1**

Jakub L. Nowak, Holger Siebert, Kai-Erik Szodry, Szymon P. Malinowski

We are grateful to the Referee #1 for the insightful comments and suggestions on our manuscript. We respond to them in detail below. The original review is given in black, our anwers in blue. The responses also mention the specific corrections which were applied to the manuscript.

**General comments**

While it is longer than most manuscripts that I review, I'm not sure that it can be substantially shortened without omitting important information.

We considered this issue before and reached the same conclusion that presenting the entire material together is of advantage for understanding the differences in turbulence character between coupled and decoupled STBL cases. Importantly, we designed most of the figures so that they fit into one column of the ACP layout. When typeset in two-column, the manuscript contains 24 pages, with last four occupied by the tables and references.

**Specific comments**

1. Lines 43–45: Could a reduction in cloud-top LW cooling due to an overrunning cloud layer at somewhat higher altitude also contribute to decoupling?

   During the day, such an overrunnig cloud layer would also reduce the solar heating of stratocumulus top. The solar heating is known to promote STBL decoupling. It is not clear to us which effect is dominant. We speculate it might depend on the height of the upper cloud and its radiative properties. During the night, the net cooling would be indeed reduced which itself favors decoupling, but on the other hand this hinders the entrainment and growth of the boundary layer. Therefore, the relative importance of those effects needs to be quantified. Unfortunately, we are not aware of the relevant studies supporting the mechanism suggested by the reviewer. Once we find such, we will update the introduction of our manuscript accordingly.

   Such mechanism was most likely not relevant for the STBL decoupling observed in flight #14. During the flight on 18 July 2017, no overlying cloud layer was reported by the scientists onboard the helicopter. In the substantial region around the operation area, the satellite products derived from MODIS onboard Aqua (NASA Worldview portal) indicate cloud

top temperature in the class of 285-290 K and cloud top height in the classs of 800-1600 m, both consistent with our observations of stratocumulus top (c.f. Fig. 6).

2. Line 106: LEGs are described as being 10 km long, but the time intervals shown on Fig. 2 seem too short at the nominal flight speed of 20 m/sec. I would prefer to see lengths and altitudes of the LEGs included in a table. Among other things, this is relevant to the question of flux sampling error (see comment further down).

The horizontal segments flown by the platform were indeed at least 10 km long. However, the manual segmentation resulted in shorter LEGs selected for the analysis as pointed out by the reviewer. In fact, LEGs are between 3.5 and 12 km long (see Table 1). This segmentation was performed in somewhat conservative manner in order to ensure that there is no potential influence of turns or pendulum-like motion of the payload on the measurement of turbulent fluctuations. This issue also relates to the next comment concerning helicopter rotor downwash.

Our description of the segmentation was incomplete with respect to the lenghts. After correction it reads:

> Segments of two types were selected from the measurement records: profiles (PROFs) and horizontal legs (LEGs). For convenience, they are ordered according to their time of execution and referred to as PROF1-PROF5 and LEG1-LEG5, for each flight. The segmentation was done manually so that the influence of sharp turns and pendulum-like motion of the payload is avoided. This resulted in the reduced length of the LEGs, between 3.5 and 12 km. LEGs were flown with TAS of 15-20 $\mathrm{m\,s^{-1}}$ and some minor displacements in vertical are unavoidable for the payload on a 170 m long rope. The mean altitudes and exact lengths are listed in Table 1. PROFs are in fact slanted with an ascent or descent rate of about 3-5 $\mathrm{m\,s^{-1}}$ and TAS $\sim$20 $\mathrm{m\,s^{-1}}$. The horizontal component of motion is necessary to avoid the downwash of the helicopter affecting wind and turbulence measurements on ACTOS.

**Table 1.** Mean altitude and length of the LEGs.

| Flight #5 | LEG5 | LEG4 | LEG3 | LEG2 | LEG1 |
|---|---|---|---|---|---|
| Height [m] | 307 | 553 | 819 | 1079 | 2018 |
| Length [km] | 5.44 | 5.51 | 7.93 | 3.94 | 6.25 |

| Flight #14 | LEG1 | LEG2 | LEG3 | LEG5 | LEG4 |
|---|---|---|---|---|---|
| Height [m] | 143 | 287 | 448 | 992 | 2021 |
| Length [km] | 8.11 | 11.92 | 7.10 | 4.79 | 3.49 |

3. The helicopter used weighs somewhere around 2000 kg and imparts substantial downward momentum and turbulent kinetic energy to the environment directly below it. In fact, rotor downwash speeds a short distance below the helicopter are probably around 30 m/sec, and the area of influence expands considerably with distance below the aircraft (albeit

with reduced velocities). With that in mind, I would have liked to see more discussion, including any relevant references, in support of the assumption that a 20 m/sec forward speed is sufficient to avoid any influence by the rotorwash on the ACTOS package suspended 150 m below the helicopter, taking into account as well that the package probably trails behind the helicopter by some distance during forward flight.

These issues are definitely worth to discuss and essential for high resolution turbulence observations with ACTOS. There are two major points to be considered:

(a) A helicopter has two completely different modes of operation (i) hovering and (ii) forward motion (and a transition phase at a true airspeed of a few meters per second which we do not consider here). During take-off the helicopter is in hovering mode and you can see (and feel) the influence of the downwash even if the helicopter is 150 m above ACTOS - this is particularly true if the wind is weak. However, on forward motion the complete rotor blade area is tilted and the downwash is deflected backwards. By the way, that is the reason why a Pitot static tube at the nose of the helicopter provides precise true airspeed even less than 2 m below the rotor blades.

(b) Any possible influence of the downwash should be visible in a power spectrum. This has been evaluated by colleagues operating a similar helicopter towed system called Helipod (Muschinski et al., 2001). They operate at 40 m/s with a 15 m rope but apply a 5-hole probe to sample turbulence. They see a sharp signal in the spectrum due to the sound waves.

This discussion with even more details has been published in the previous publication about ACTOS (Siebert et al., 2006) which has been cited at the beginning of the instrumental part (Sec 2.2.) of our manuscript. Therefore, we suggest to avoid a repetition of this discussion here but included a sentence for interested readers:

> More details about measuring turbulence below a helicopter can be found in Siebert et al. (2006).

4. I believe there should be explicit discussion of sampling error, and its relationship to flight leg length, in connection with the turbulent flux measurements. One newly published paper that seems relevant is Petty, G. W.: Sampling error in aircraft flux measurements based on a high-resolution large eddy simulation of the marine boundary layer, Atmos. Meas. Tech., 14, 1959–1976, https://doi.org/10.5194/amt-14-1959-2021, 2021.

We have already performed a comprehensive analysis of sampling error using the methods given by Lenschow et al. (1994), hereafter L94, with respect to our LEG measurements of turbulent fluxes as well as turbulent variances. However, taking into account the length of the manuscript, we decided to only show the standard deviation among the relevant values derived separately for seven subsegments (Std7, see sec. 4) because the same method can be applied to other variables in our work, in particular turbulence parameters (e.g. dissipation rate or anisotropy ratios) for which the rigorous and practical formulas for systematic/random errors are not available. Moreover, we found that Std7 is of the same order as random error (L94ran), exceeding it in most of the cases, while systematic error (L94sys) is significantly smaller. The detailed analysis is intended to be covered in the doctoral dissertation of Jakub L. Nowak. Here, we briefly describe the procedure and present the results.

For each variable $x$ out of $u$, $\theta_v$, $q_v$, $T$, integral lengthscales $L_x$, $L_{wx}$, $L_f$ (corresponding to autocovariance of $x'$, covariance of $w'$ and $x'$, autocovariance of the product $w'x'$, respectively) were estimated with the procedure described in sec. 4.5 of our manuscript. Accordingly, we calculated correlation coefficient $r_{wx}$ of $w'$ and $x'$. Those values are listed in Tables 2 and 3 for flights #5 and #14, respectively.

For variances, systematic error was estimated using Eq. (14) while random error using Eq. (36) of L94. In case of the third moment of vertical velocity, the coefficient $a$ was found by solving their Eq. (20) and then its value was applied to estimate systematic error according to their Eq. (21). Random error of $\langle w'^3 \rangle$ was estimated according to Eq. (B40) of Lenschow et al. (1993). The errors are compared with Std7 in Tables 4 and 5 for flights #5 and #14, respectively.

For fluxes, systematic error was estimated with Eq. (30) and random error with Eq. (48) of L94. The latter is also the equation upon which Petty (2021) builds his analysis. He proved this equation to be very accurate at predicting random error for flight tracks of the length relevant for our LEGs. The errors are compared with the subsegment variability in Tables 6 and 7 for flights #5 and #14, respectively.

Following the reviewer's request we briefly discuss the issue of the sampling errors in sec. 4.1 and 4.2 of the manuscript:

> The accuracy of the results is limited by the length of the LEGs. Based on the estimates obtained with the methods of Lenschow et al. (1994), in the boundary layer the variances are subject to the systematic sampling error of about 5 % and the random error of about 20 %. In case of $\langle w'^3 \rangle$, those errors are accordingly larger (order of 10 % and 100 %, respectively, unless $\langle w'^3 \rangle$ is not very close to zero). Importantly, in the plots we provide the variability among subsegments because it can be estimated for other variables as well, in particular turbulence parameters, and it was found to be of the same order as the total sampling error, in most cases larger than it (not shown).

> Similarly to variances, the accuracy of the fluxes obtained with eddy correlation is limited by the length of the LEGs. In the boundary layer, systematic error was estimated for about 5-10 % while random error for about 50 % (Lenschow et al., 1994), unless the flux does not vanish. The subsegment variability (marked with errorbars in the plots) is in most cases larger than the total sampling error.

**References**

Lenschow, D. H., Mann, J., and Kristensen, L.: How Long is Long Enough when Measuring Fluxes and Other Turbulence Statistics?, Tech. rep., University Corporation for Atmospheric Research, 1993.

Lenschow, D. H., Mann, J., and Kristensen, L.: How long is long enough when measuring fluxes and other turbulence statistics?, Journal of Atmospheric and Oceanic Technology, 11, 661–673, https://doi.org/10.1175/1520-0426(1994)011<0661:HLILEW>2.0.CO;2, 1994.

Muschinski, A., Frehlich, R., Jensen, M., Hugo, R., Hoff, A., Eaton, F., and Balsley, B.: Fine-scale measurements of turbulence in the lower troposphere: An intercomparison between a kit-and balloon-borne, and a helicopter-borne measurement system, Boundary-Layer Meteorology, 98, 219–250, https://doi.org/10.1023/A:1026520618624, 2001.

Petty, G. W.: Sampling error in aircraft flux measurements based on a high-resolution large eddy simulation of the marine boundary layer, Atmospheric Measurement Techniques, 14, 1959–1976, https://doi.org/10.5194/amt-14-1959-2021, 2021.

Siebert, H., Franke, H., Lehmann, K., Maser, R., Saw, E. W., Schell, D., Shaw, R. A., and Wendisch, M.: Probing finescale dynamics and microphysics of clouds with helicopter-borne measurements, Bulletin of the American Meteorological Society, 87, 1727–1738, https://doi.org/10.1175/BAMS-87-12-1727, 2006.

**Table 2.** Integral scales and correlations in flight #5.

| | Variable | LEG5 | LEG4 | LEG3 | LEG2 | LEG1 |
|---|---|---|---|---|---|---|
| | Height [m] | 307 | 553 | 819 | 1079 | 2018 |
| | Length [km] | 5.44 | 5.51 | 7.93 | 3.94 | 6.25 |
| $w'$ | $L_w$ [m] | 146 | 120 | 97 | 112 | 64 |
| $u'$ | $L_u$ [m] | 113 | 154 | 179 | 101 | 112 |
| | $L_{wu}$ [m] | 520 | 54 | 31 | 185 | 182 |
| | $L_f$ [m] | 80 | 83 | 39 | 59 | 46 |
| | $r_{wu}$ [m] | -0.26 | -0.20 | -0.11 | 0.21 | -0.31 |
| $\theta'_v$ | $L_{\theta_v}$ [m] | 108 | 110 | 172 | 79 | 117 |
| | $L_{w\theta_v}$ [m] | 353 | 655 | 255 | 25 | 50 |
| | $L_f$ [m] | 48 | 59 | 56 | 37 | 12 |
| | $r_{w\theta_v}$ [m] | 0.30 | -0.08 | 0.46 | -0.09 | -0.16 |
| $q'_v$ | $L_{q_v}$ [m] | 160 | 136 | 94 | 120 | 318 |
| | $L_{wq_v}$ [m] | 157 | 82 | 200 | NaN | 152 |
| | $L_f$ [m] | 48 | 90 | 43 | 88 | 18 |
| | $r_{wq_v}$ [m] | 0.56 | 0.06 | 0.54 | -0.00 | 0.09 |
| $T'$ | $L_T$ [m] | 76 | 129 | 204 | 108 | 250 |
| | $L_{wT}$ [m] | 156 | 177 | 238 | 178 | 59 |
| | $L_f$ [m] | 21 | 46 | 51 | 30 | 27 |
| | $r_{wT}$ [m] | 0.27 | 0.29 | 0.55 | -0.12 | 0.14 |

**Table 3.** Integral scales and correlations in flight #14.

| | Variable | LEG1 | LEG2 | LEG3 | LEG5 | LEG4 |
|---|---|---|---|---|---|---|
| | Height [m] | 143 | 287 | 448 | 992 | 2021 |
| | Length [km] | 8.11 | 11.92 | 7.10 | 4.79 | 3.49 |
| $w'$ | $L_w$ [m] | 53 | 86 | 74 | 35 | 249 |
| $u'$ | $L_u$ [m] | 103 | 111 | 109 | 105 | 180 |
| | $L_{wu}$ [m] | 54 | 88 | 63 | 64 | 6 |
| | $L_f$ [m] | 33 | 46 | 29 | 28 | 28 |
| | $r_{wu}$ [m] | 0.10 | -0.18 | -0.13 | 0.14 | -0.00 |
| $\theta'_v$ | $L_{\theta_v}$ [m] | 85 | 50 | 153 | 62 | 115 |
| | $L_{w\theta_v}$ [m] | 69 | 6 | 596 | 107 | 21 |
| | $L_f$ [m] | 19 | 13 | 71 | 23 | 12 |
| | $r_{w\theta_v}$ [m] | 0.29 | 0.03 | -0.10 | 0.28 | -0.16 |
| $q'_v$ | $L_{q_v}$ [m] | 79 | 119 | 117 | 82 | 74 |
| | $L_{wq_v}$ [m] | 69 | 89 | 489 | 244 | 160 |
| | $L_f$ [m] | 34 | 60 | 62 | 35 | 25 |
| | $r_{wq_v}$ [m] | 0.44 | 0.42 | 0.07 | 0.04 | 0.11 |
| $T'$ | $L_T$ [m] | 52 | 76 | 127 | 49 | 28 |
| | $L_{wT}$ [m] | 33 | 2 | 148 | 130 | 489 |
| | $L_f$ [m] | 29 | 21 | 20 | 23 | 14 |
| | $r_{wT}$ [m] | 0.24 | 0.01 | 0.17 | 0.16 | 0.28 |

**Table 4.** Statistical errors of the LEG-derived moments in flight #5 (coupled STBL): standard deviation among subsegments (Std7), systematic and random errors according to L94 (L94sys and L94ran).

| Variable | | LEG5 | | LEG4 | | LEG3 | | LEG2 | | LEG1 | |
|---|---|---|---|---|---|---|---|---|---|---|---|
| Height [m] | | 307 | | 553 | | 819 | | 1079 | | 2018 | |
| $\langle w'^2 \rangle$ | [m² s⁻²,%] | 0.212 | | 0.104 | | 0.162 | | 0.004 | | 0.006 | |
| | Std7 | 0.087 | 41 | 0.018 | 17 | 0.056 | 35 | 0.001 | 23 | 0.002 | 29 |
| | L94sys | 0.011 | 5.4 | 0.005 | 4.4 | 0.004 | 2.4 | 0.000 | 5.7 | 0.000 | 2.0 |
| | L94ran | 0.049 | 23 | 0.022 | 21 | 0.025 | 16 | 0.001 | 24 | 0.001 | 14 |
| $\langle w'^3 \rangle$ | [10⁻²m³ s⁻³,%] | 1.69 | | 0.06 | | 0.23 | | -0.01 | | -0.02 | |
| | Std7 | 2.20 | 130 | 1.55 | 2658 | 2.12 | 908 | 0.02 | 182 | 0.02 | 108 |
| | L94sys | 0.23 | 13.4 | 0.01 | 10.9 | 0.01 | 6.1 | 0.00 | 14.3 | 0.00 | 5.1 |
| | L94ran | 3.38 | 200 | 0.99 | 1699 | 1.45 | 619 | 0.01 | 130 | 0.01 | 74 |
| $\langle u'^2 \rangle$ | [m² s⁻²,%] | 0.34 | | 0.27 | | 0.20 | | 0.37 | | 0.20 | |
| | Std7 | 0.10 | 29 | 0.09 | 32 | 0.03 | 13 | 0.13 | 34 | 0.10 | 49 |
| | L94sys | 0.01 | 4.2 | 0.02 | 5.6 | 0.01 | 4.5 | 0.02 | 5.1 | 0.01 | 3.6 |
| | L94ran | 0.07 | 20 | 0.06 | 24 | 0.04 | 21 | 0.08 | 23 | 0.04 | 19 |
| $\langle q_v'^2 \rangle$ | [10⁻³g² kg⁻²,%] | 5.4 | | 7.8 | | 26.4 | | 1.4 | | 0.0 | |
| | Std7 | 1.0 | 18 | 1.6 | 21 | 17.1 | 65 | 1.3 | 91 | 0.0 | 18 |
| | L94sys | 0.3 | 5.9 | 0.4 | 4.9 | 0.6 | 2.4 | 0.1 | 6.1 | 0.0 | 10.2 |
| | L94ran | 1.3 | 24 | 1.7 | 22 | 4.1 | 15 | 0.3 | 25 | 0.0 | 32 |
| $\langle T'^2 \rangle$ | [10⁻³K²,%] | 3.3 | | 7.5 | | 18.4 | | 7.3 | | 7.8 | |
| | Std7 | 0.9 | 26 | 3.0 | 39 | 8.0 | 43 | 6.3 | 86 | 3.7 | 47 |
| | L94sys | 0.1 | 2.8 | 0.4 | 4.7 | 0.9 | 5.1 | 0.4 | 5.5 | 0.6 | 8.0 |
| | L94ran | 0.6 | 17 | 1.6 | 22 | 4.2 | 23 | 1.7 | 23 | 2.2 | 28 |

**Table 5.** Statistical errors of the LEG-derived moments in flight #14 (decoupled STBL): standard deviation among subsegments (Std7), systematic and random errors according to L94 (L94sys and L94ran).

| Variable | | LEG5 | | LEG4 | | LEG3 | | LEG2 | | LEG1 | |
|---|---|---|---|---|---|---|---|---|---|---|---|
| Height [m] | | 143 | | 287 | | 448 | | 992 | | 2021 | |
| $\langle w'^2 \rangle$ | [m$^2$ s$^{-2}$,%] | 0.106 | | 0.076 | | 0.047 | | 0.054 | | 0.004 | |
| | Std7 | 0.036 | 34 | 0.027 | 36 | 0.014 | 30 | 0.012 | 22 | 0.002 | 35 |
| | L94sys | 0.001 | 1.3 | 0.001 | 1.5 | 0.001 | 2.1 | 0.001 | 1.5 | 0.001 | 14.3 |
| | L94ran | 0.012 | 11 | 0.009 | 12 | 0.007 | 14 | 0.007 | 12 | 0.002 | 38 |
| $\langle w'^3 \rangle$ | [10$^{-2}$m$^3$ s$^{-3}$,%] | 0.49 | | 1.34 | | -0.21 | | -0.47 | | -0.01 | |
| | Std7 | 1.33 | 272 | 0.88 | 66 | 0.53 | 247 | 0.28 | 59 | 0.02 | 287 |
| | L94sys | 0.02 | 3.3 | 0.05 | 3.6 | 0.01 | 5.2 | 0.02 | 3.7 | 0.00 | 35.7 |
| | L94ran | 0.58 | 119 | 0.59 | 44 | 0.23 | 105 | 0.27 | 57 | 0.02 | 295 |
| $\langle u'^2 \rangle$ | [m$^2$ s$^{-2}$,%] | 0.21 | | 0.14 | | 0.27 | | 0.19 | | 0.05 | |
| | Std7 | 0.11 | 53 | 0.04 | 25 | 0.05 | 18 | 0.05 | 27 | 0.01 | 32 |
| | L94sys | 0.01 | 2.5 | 0.00 | 1.9 | 0.01 | 3.1 | 0.01 | 4.4 | 0.00 | 10.3 |
| | L94ran | 0.03 | 16 | 0.02 | 14 | 0.05 | 18 | 0.04 | 21 | 0.01 | 32 |
| $\langle q_v'^2 \rangle$ | [10$^{-3}$g$^2$ kg$^{-2}$,%] | 27.1 | | 12.0 | | 44.7 | | 31.7 | | 1.2 | |
| | Std7 | 6.1 | 23 | 3.5 | 30 | 21.0 | 47 | 9.8 | 31 | 0.4 | 32 |
| | L94sys | 0.5 | 1.9 | 0.2 | 2.0 | 1.5 | 3.3 | 1.1 | 3.4 | 0.1 | 4.3 |
| | L94ran | 3.8 | 14 | 1.7 | 14 | 8.1 | 18 | 5.9 | 19 | 0.3 | 21 |
| $\langle T'^2 \rangle$ | [10$^{-3}$K$^2$,%] | 6.7 | | 5.9 | | 11.0 | | 7.7 | | 4.9 | |
| | Std7 | 0.6 | 9 | 1.1 | 18 | 3.4 | 31 | 3.0 | 39 | 0.8 | 17 |
| | L94sys | 0.1 | 1.3 | 0.1 | 1.3 | 0.4 | 3.6 | 0.2 | 2.0 | 0.1 | 1.6 |
| | L94ran | 0.8 | 11 | 0.7 | 11 | 2.1 | 19 | 1.1 | 14 | 0.6 | 13 |

**Table 6.** Statistical errors of the LEG-derived fluxes in flight #5 (coupled STBL): standard deviation among subsegments (Std7), systematic and random errors according to L94 (L94sys and L94ran).

| | Variable | LEG5 | | LEG4 | | LEG3 | | LEG2 | | LEG1 | |
|---|---|---|---|---|---|---|---|---|---|---|---|
| | Height [m] | 307 | | 553 | | 819 | | 1079 | | 2018 | |
| $B$ | $[10^{-4}\,\mathrm{m}^2\,\mathrm{s}^{-3},\%]$ | 1.1 | | 0.1 | | 8.0 | | -0.1 | | -0.5 | |
| | Std7 | 2.7 | 243 | 2.8 | 3055 | 4.4 | 55 | 0.2 | 170 | 0.4 | 84 |
| | L94sys | 0.1 | 12.1 | 0.0 | 20.9 | 0.5 | 6.2 | 0.0 | 1.2 | 0.0 | 1.6 |
| | L94ran | 0.5 | 46 | 0.2 | 188 | 2.3 | 28 | 0.2 | 149 | 0.2 | 40 |
| $S$ | $[10^{-4}\,\mathrm{m}^2\,\mathrm{s}^{-3},\%]$ | 2.7 | | 1.6 | | 1.7 | | 0.9 | | 1.0 | |
| | Std7 | 5.1 | 272 | 2.0 | 235 | 1.4 | 110 | 0.6 | 68 | 1.1 | 124 |
| | L94sys | 0.5 | 17.3 | 0.0 | 1.9 | 0.0 | 0.8 | 0.1 | 8.9 | 0.1 | 5.6 |
| | L94ran | 1.9 | 69 | 1.4 | 87 | 1.5 | 88 | 0.8 | 86 | 0.4 | 41 |
| $Q_s$ | $[\mathrm{W\,m}^{-2},\%]$ | 4.0 | | 7.2 | | 38.0 | | -0.6 | | 1.2 | |
| | Std7 | 2.7 | 66 | 6.6 | 91 | 20.5 | 54 | 0.4 | 70 | 0.8 | 66 |
| | L94sys | 0.2 | 5.6 | 0.4 | 6.2 | 2.2 | 5.8 | 0.0 | 8.6 | 0.0 | 1.9 |
| | L94ran | 1.3 | 33 | 3.3 | 46 | 8.9 | 23 | 0.6 | 104 | 0.8 | 67 |
| $Q_l$ | $[\mathrm{W\,m}^{-2},\%]$ | 50.4 | | 5.0 | | 104.6 | | 0.1 | | 0.0 | |
| | Std7 | 22.9 | 45 | 22.6 | 456 | 77.7 | 74 | 1.0 | 975 | 0.0 | 75 |
| | L94sys | 2.8 | 5.6 | 0.1 | 2.9 | 5.1 | 4.9 | NaN | NaN | 0.0 | 4.7 |
| | L94ran | 13.7 | 27 | 13.9 | 281 | 22.9 | 22 | 8.3 | 7869 | 0.0 | 81 |

**Table 7.** Statistical errors of the LEG-derived fluxes in flight #14 (decoupled STBL): standard deviation among subsegments (Std7), systematic and random errors according to L94 (L94sys and L94ran).

| | Variable | LEG1 | | LEG2 | | LEG3 | | LEG5 | | LEG4 | |
|---|---|---|---|---|---|---|---|---|---|---|---|
| | Height [m] | 143 | | 287 | | 448 | | 992 | | 2021 | |
| $B$ | $[10^{-4}\,\mathrm{m^2\,s^{-3}},\%]$ | 2.7 | | 0.3 | | -0.3 | | 2.6 | | -0.2 | |
| | Std7 | 1.5 | 57 | 0.5 | 185 | 1.7 | 533 | 0.9 | 36 | 0.2 | 80 |
| | L94sys | 0.0 | 1.7 | 0.0 | 0.1 | 0.1 | 15.4 | 0.1 | 4.4 | 0.0 | 1.2 |
| | L94ran | 0.6 | 24 | 0.5 | 183 | 0.5 | 138 | 0.9 | 37 | 0.1 | 53 |
| $S$ | $[10^{-4}\,\mathrm{m^2\,s^{-3}},\%]$ | 0.7 | | 2.3 | | 1.5 | | 1.7 | | 0.1 | |
| | Std7 | 0.9 | 137 | 0.9 | 67 | 1.3 | 87 | 1.6 | 125 | 0.1 | 175 |
| | L94sys | 0.0 | 1.3 | 0.0 | 1.5 | 0.0 | 1.8 | 0.0 | 2.6 | 0.0 | 0.3 |
| | L94ran | 0.7 | 94 | 1.2 | 51 | 1.1 | 70 | 1.4 | 79 | 4.7 | 4830 |
| $Q_s$ | $[\mathrm{W\,m^{-2}},\%]$ | 8.9 | | 1.6 | | 3.6 | | 2.5 | | 0.4 | |
| | Std7 | 3.3 | 37 | 4.1 | 252 | 2.7 | 75 | 3.8 | 154 | 0.6 | 177 |
| | L94sys | 0.1 | 0.8 | 0.0 | 0.0 | 0.1 | 4.1 | 0.1 | 5.3 | 0.1 | 24.1 |
| | L94ran | 3.3 | 37 | 14.3 | 884 | 1.6 | 46 | 1.6 | 63 | 0.1 | 33 |
| $Q_l$ | $[\mathrm{W\,m^{-2}},\%]$ | 76.7 | | 39.3 | | 6.8 | | 11.5 | | 0.8 | |
| | Std7 | 36.1 | 47 | 28.1 | 72 | 43.5 | 639 | 23.3 | 203 | 0.7 | 93 |
| | L94sys | 1.3 | 1.7 | 0.6 | 1.5 | 0.9 | 12.8 | 1.1 | 9.7 | 0.1 | 8.8 |
| | L94ran | 17.5 | 23 | 10.1 | 26 | 13.4 | 197 | 33.8 | 294 | 0.8 | 107 |

---

## Author Comment (AC2)

**Authors' Response to the Anonymous Referee #2**

Jakub L. Nowak, Holger Siebert, Kai-Erik Szodry, Szymon P. Malinowski

We are grateful to the Referee #2 for the insightful comments and suggestions on our manuscript. We respond to them in detail below. The original review is given in black, our answers in blue. The responses also mention the specific corrections which were applied to the manuscript.

**General comments**

1. The paper provides interesting insights into the stratification and the turbulent properties of coupled (CP) and decoupled (DCP) marine stratocumulus-topped boundary layers. It would be interesting to have additional CP/DCP cases to investigate whether the same pattern is also observed in other cases. Are you planning to extend the analysis to more cases? It would be in particular interesting to have more data in the stratocumulus layer.

   We are grateful for the comment, which partly coincides with the first comment from the reviewer #4 (Ian Brooks). We will consider extending this analysis to other available cases, though, as usual this will depend on the available manpower. For this manuscript, we have chosen to focus on details rather than an extended data set. However, we also believe that extending similar analyses to more flights will make the results more robust.

   On the other hand, the ACORES data alone might not be sufficient to provide statistically sound conclusions and we consider extending the analyzed dataset with the available data from other field experiments. In total, there were 17 research flights during the ACORES (Siebert et al., 2021, Table 5). Five of them correspond to clear-sky conditions, four to already dissipating or not yet developed stratocumulus clouds which limits the true STBL observations to 8 flights. Each flight lasted up to about 2 hours. This flight time was always disposed between sampling the cloud top structure and the boundary layer itself.

2. In Sect. 2.2, you introduce the different instruments that have been used in the study. I would like to see further discussion regarding the uncertainties of the measurements. Currently, uncertainties are not discussed and the error bars only include the variability of the data. This makes it difficult to assess the conditions inside the different sublayers in the CP and DCP case (e.g., add error bars in Table A1 and A2).

   In our study, we present a great variety of turbulence parameters. We suppose it is impossible to apply one universal and rigorous approach of error treatment for such different variables. For this reason, we decided to report subsegment variability because it can be evaluated regardless of the details of a particular derivation method. On the other hand, we

agree with the reviewers that the issue of uncertainties was not comprehensively addressed. In order to improve this, we complemented the manuscript with a few additional paragraphs discussing the aspects which we find the most relevant for the derived turbulence parameters.

We described the capabilities of the instruments onboard ACTOS in sec. 2.2.

> The standard deviations due to uncorrelated noise for sonic measurements are $0.02 \ \mathrm{m\,s^{-1}}$ for wind and $0.02$ K for virtual temperature (Siebert and Muschinski, 2001). The PT100 was calibrated prior to the campaign in a thermostated water tank using the Greisinger GMH 3750 reference thermometer which provides accuracy better than 0.05 K. The UFT was calibrated for each flight separately against the PT100. For the UFT records, the standard deviation due to uncorrelated noise is 4 mK (Siebert et al., 2003). The hygrometer provides $q_v$ with a noise floor of about $0.005 \ \mathrm{g\,kg^{-1}}$. This instrument was verified to agree well with a few hygrometers of different types and operate satisfactorily on the helicopter-towed system Helipod by Lampert et al. (2018). The PVM-100A measures $q_l$ with the accuracy of 5 % and its noise floor was estimated by Siebert et al. (2003) for about $0.001 \ \mathrm{g\,kg^{-1}}$. The exact sensitivity depends to some extent on droplet size distribution, see Wendisch et al. (2002) for details. For a more general discussion of the instrumentation on the ACTOS platform see Siebert et al. (2006a).

We discussed the sampling errors (systematic and random) for turbulent moments (variances, TKE, $\langle w'^3 \rangle$) estimated according to Lenschow et al. (1993, 1994) in sec. 4.1. The detailed procedure with all the specific values is delineated in our response to the Anonymous Reviewer #1.

> The accuracy of the results is severely limited by the length of the LEGs. Based on the methods of Lenschow et al. (1994), in the boundary layer the variances are subject to the systematic sampling error of about 5 % and the random sampling error of about 20 %. In the case of $\langle w'^3 \rangle$, those errors are accordingly larger (order of 10 % and 100 %, respectively, unless $\langle w'^3 \rangle$ is not very close to zero). Importantly, in the plots we provide the variability among subsegments which was found to be of the same order as the total sampling error, in most cases larger than it.

Similarly, we discussed the sampling errors for turbulent fluxes in sec. 4.2. Again, please see the response given to the Anonymous Reviewer #1 to find the tables presenting all the individual errors.

> Similarly to variances, the accuracy of the fluxes obtained with the method of eddy correlation is limited by the length of the LEGs. In the boundary layer, the systematic sampling error was estimated for about 5-10 % while the random sampling error for about 50 % (Lenschow et al., 1994), unless the flux does not vanish. The subsegment variability (marked with errorbars in the plots) is in most cases larger than the total sampling error.

We estimated the uncertainties of the derived dissipation rates and SFC/PSD slopes due to random errors in sec. 4.3.3. In addition, we proposed alternative methods of a rather qualitative assessment of the reliability of the results.

In order to roughly estimate the uncertainties of the results, we used the random errors of the fitted parameters (computed with a standard method from least-squares fit residuals). The random error of 'instantaneous' (calculated in 2 s windows and serving for the derivation of the profiles) dissipation rate equals ∼50 % in the boundary layer and ∼150 % in the FT. The error of the LEG-derived $\epsilon$ is ∼30 % for longitudinal component and ∼15 % for vertical component in the boundary layer while ∼150 % for both components in the FT. The random error of the fitted slopes is ∼0.04 for $s$ and ∼0.16 for $p$ corresponding to the 'instantaneous' estimations while ∼0.02 in the case of both LEG-derived slopes. Notwithstanding, the given values represent the uncertainties due to the random errors of the fit only. The reliability of the derived dissipation rates can be also assessed by comparing the results of the two derivation methods, by comparing the fitted SFC and PSD slopes with their theoretical values or using the deviation of the computed correlation coefficients from unity.

The uncertainties of further quantities derived from dissipation rate can be estimated by the method of error propagation. Additionally, we referred to the previous works to argue that the analysis of spectral anisotropy (sec. 4.4 and 5.4) is justified taking into account the quality of our data. Our data is sufficient for the analysis of the inertial range anisotropy as Siebert and Muschinski (2001) demonstrated that the spectra of velocity fluctuations measured with an earlier version of our ultrasonic anemometer-thermometer in a considerably turbulent environment follow closely the expected 5/3 power law, a flattening is observed only at frequencies larger than 30 Hz and the ratio of the transverse and longitudinal spectra equals 4/3, as predicted for isotropic turbulence.

3. The paper contains many abbreviations (e.g., for the different sublayers). I understand that it makes sense to introduce these abbreviations, as they are frequently used in the paper. However, for the readers that are not familiar with these abbreviations, it can be hard to remember the definition of the different acronyms and to follow the text. Please check if all the abbreviations are necessary. For example, "ENA" and "CTEI" are only used 2-3 times and could be removed. Furthermore, I would suggest including the abbreviations of the different sublayers in the figures, in order to make it easier for the reader to identify them (see specific comments).

Following the suggestion of this and other reviewers we reduced the number of acronyms by replacing them with the corresponding expanded expressions, in particular those which were not used frequently in the text. This group includes: SC, BL, ENA, TAS, CTEI, J11, WB04, YA00, CP, DCP. The last two were shortened to C and D, respectively, and only kept in sec. 6 to order the list of conclusions. We prefer to keep the acronyms of the following types:

- denoting the sublayers of the atmosphere: STBL, SML, TSL, SBL, SCL, EIL, FT, because they are used very frequently in the text as well as in tables and figures,

- denoting our flight segments: PROF, LEG, for the same reason,

- commonly used abbreviations: TKE, LCL, SFC, PSD, because we expect them to be familiar to the readers,

- names of instruments or platforms: ACTOS, SMART-HELIOS, MODIS, GPS, for the same reason.

Moreover, we added the expanded names of the sublayers to the headings of Tables A1 and A2.

The abbreviations denoting different sublayers were added to Figs. 5, 6, 9, 10, 11, 12, 13, 14, 15, 16, 19, 20 as suggested.

**Specific comments**

4. Page 5, caption Fig. 1 and caption Fig.3: Please add date and time of satellite image.

   The captions were corrected as suggested. Fig. 1. shows the imagery acquired on 8 July 2017 at 15:45 UTC, Fig. 3. on 18 July 2017 at 14:43 UTC.

5. Page 5, Fig. 2 and Fig. 4: In Fig. 2 and Fig. 4 you show the time series of the ACTOS altitude. I think it would be beneficial to include more information regarding the cloud/BL structure. E.g., At what altitude is the cloud top/cloud base? You could indicated the different sublayers on the right side of the plot. Furthermore, you often refer to the different profiles (PROFs 1-5) and legs (LEGs 1-5) throughout the paper. You could consider adding the labels of the profiles and legs on top of the plot. In addition, the line style of the profiles is not evident in the figure due to the low contrast between the black line in the background and the black dotted line. I would suggest to remove the black line in the background or to change the color to get a better contrast.

   The figures were modified according to the suggestions. The labels denoting PROFs and LEGs were added at the top. The ordering of LEGs was changes into LEGX where X stands for mean altitude (m a.s.l.), following the request of another reviewer. The grey line illustrating altitude profile for the whole flight was plotted only outside the colored segments to improve the visibility of the black line used within the segments. Instead of sublayer labels, the individual penetrations (determined manually) of the EIL top, SCL top (=EIL base), SCL base (=SBL top), TSL top (=SBL bottom) and TSL base were indicated in the altitude profile with additional symbols.

[Figure]

**Figure 2 corrected.** ACTOS altitude in flight #5 with marked selected profiles and horizontal legs. PROFs are ordered chronologically, LEGs are ordered according to their altitude. Line styles of the PROFs are consistent with the figures in following sections Altitude ranges corresponding to PROF2-PROF5 of this flight do not overlap and are all marked with dotted lines. Dots indicate the penetrations of the boundaries of the specific sublayers described in sec. 3.

[Figure]

**Figure 4 corrected.** As in Fig. 2 but for flight #14. Line styles of the PROFs are consistent with the figures in following sections. PROF1-PROF3 are all marked with dotted lines because their altitude ranges do not overlap.

6. Page 6, equation 1: You defined 'ql' as the liquid water content on page 4, line 103. The liquid water content is usually defined as mass of liquid water per volume of air (i.e. g m -3 ). However, in equation 1 the liquid water mixing ratio (i.e. mass of liquid water per unit mass of air) should be used and not the liquid water content (see Betts 1973 or the following link: https://glossary.ametsoc.org/wiki/Liquid_water_potential_temperature). Please review your definition of 'ql' in the manuscript.

In our calculations and throughout the manuscript $q_l$ denotes liquid water mass fraction, i.e. mass of liquid water in a unit mass of moist liquid-ladden air. Its units are $\mathrm{g\,kg^{-1}}$. It is consistent with Eq. (14) of Betts (1973) and with $q_t = q_v + q_l$ being a conservative quantity. Such definition is related to liquid water content (mass of liquid water per unit volume of air) $\rho_l$ and to liquid water mixing ratio (mass of liquid water per unit mass of dry air) $r_l$ as the following:

$$q_l = \frac{\rho_l}{\rho_d + \rho_v + \rho_l} = \frac{r_l}{1 + r_v + r_l}, \quad r_l = \frac{\rho_l}{\rho_d} \tag{1}$$

where $\rho_d$ is density of dry air, $\rho_v$ is density of water vapor and $r_v$ is water vapor mixing ratio. We corrected the erroneous definition in the text.

... liquid water mass fraction $q_l$ determined with the Particle Volume Meter ...

7. Page 7, line 150: According to J11, 'qt' should be the total water mixing ratio, which is defined by the sum of the liquid water mixing ratio and the water vapor mixing ratio (see also comment 6). Please review your definition of 'qt' in the manuscript.

In our calculations and throughout the manuscript $q_t$ denotes total water mass fraction, i.e. the total mass of liquid water and water vapor per unit mass of moist liquid-ladden air (see also the answer to comment 6 above):

$$q_t = q_v + q_l = \frac{\rho_v + \rho_l}{\rho_d + \rho_v + \rho_l} = \frac{r_t}{1 + r_t}, \quad r_t = r_v + r_l, \quad r_v = \frac{\rho_v}{\rho_d} \tag{2}$$

where $q_v$ is water vapor mass fraction (specific humidity), $q_l$ is liquid water mass fraction, $\rho_d$ is density of dry air, $\rho_v$ is density of water vapor, $\rho_l$ is liquid water content. $r_v$ is water vapor mixing ratio, $r_l$ is liquid water mixing ratio and $r_t$ is total water mixing ratio. Insofar, we indeed used slightly different criterion than J11 who applied total water mixing

ratio $r_t$ instead of our total water mass fraction $q_t$. This difference do not affect the conclusions reached with the use of the criterion because approximately $q_t \approx r_t$. This controversy was briefly explained in the text.

> The first criterion of Jones et al. (2011) involves the differences of $\theta_l$ and total water mixing ratio between the uppermost and the lowermost quarters of the boundary layer (instead of the latter quantity, we used our total water mass fraction $q_t = q_l + q_v$ which does not influence the conclusions because those two measures are approximately equal).

In contrast to the criterion of Jones et al. (2011), in the one of Yin and Albrecht (2000) we did use the water vapor mixing ratio $r_v$ following those authors literally because there is derivative of this quantity involved (i.e. in general, under some conditions small discrepancies might affect the result).

8. Page 9, Fig. 5 and Fig. 6: As mentioned already in the general comment section, it is hard to remember the abbreviations of the different sublayers. In order to make it easier for the reader to follow and identify the different sublayers, I would suggest adding the abbreviations of the sublayers (color shaded areas) on the right side of the subplots (for all figures of this type; i.e. Fig. 5, 6, 9-16, 19-20). Furthermore, I would plot the lines on top of the shaded area to avoid any change in the line color (for example for LCL, qv).

The acronyms denoting different sublayers were added to Figs. 5, 6, 9, 10, 11, 12, 13, 14, 15, 16, 19, 20 as suggested. Moreover, their full names were given in the tables in the appendix. We appreciate recognizing the issue with colors while plotting the shaded areas. We changed the order of drawing in our routine as suggested.

9. Page 9, line 207: So are both the upper and the lower BL portion internally mixed? If yes, you could change the structure of the sentence as follows: "This suggests that both the upper and lower BL portion are internally mixed."

Yes, they are. We corrected the sentence as suggested.

10. Page 14, line 113: You applied a moving window of 2 s to the profiles. How was the moving window of 2 s determined? Did you conduct sensitivity tests with different time windows?

In a few previous studies which utilized the same type of data, the window of 1 s was proven to operate satisfactorily while deriving the instantaneous dissipation rate (e.g. Siebert et al., 2006b; Katzwinkel et al., 2012). The choice of such window length by Siebert et al. (2006b) followed from their own sensitivity tests and the works of Muschinski et al. (2004); Frehlich et al. (2004). Because we determine not only the dissipation rate but also the slope of the SFC or the PSD in the inertial range, we decided to increase the window to 2 s so that the linear fit covers the considerable portion of the inertial range (0.4-40 m) and the sufficient number of logarithmically equidistant resampled points (eight per decade, see sec. 4.3 of the manuscript). We did not conduct additional strict sensitivity tests within the present study. The appropriate explanatory comment was added to the text.

> Our approach follows earlier studies which determined the instantaneous dissipation rate utilizing the same type of data as ours (Siebert et al., 2006b; Katzwinkel et al., 2012). Siebert et al. (2006b) have chosen

the window of 1 s based on their sensitivity tests and the arguments provided by Frehlich et al. (2004) and Muschinski et al. (2004). Because we derive not only $\epsilon$ but also the slopes and correlations, we increased the window to 2 s so that the linear fit covers considerable portion of the inertial range and the sufficient number of logarithmically equidistant resampled points (see sec. 4.3.1).

11. Page 20, Fig.13: Why is there such a large discrepancy between some of the PROFs and LEGs properties (e.g., $s_u$, $R_u$, $R_w$ ) in the FTL?

First, the free troposphere is not expected to be turbulent. Under such conditions, the assumptions of Kolmogorov theory exploited in both the methods of dissipation rate derivation are far from being fulfilled. Structure functions and power spectra are not guaranteed to follow any specific scaling. The inertial range cannot be defined.

Second, even in the STBL the results on small-scale turbulence, including $\epsilon$, $s$ and $p$, should not be compared between PROFs and LEGs in a straightforward way. They are representative for small and large fluid volumes, respectively. Note that the climb rate of the helicopter is much higher than of a typical research aircraft (c.f. Siebert et al., 2021, sidebar "Profiling with aircraft and helicopter"). Also, horizontal segments may cover various air volumes differing in turbulence intensity and its properties, e.g. dissipation rate or inertial range scaling. According to the refined Kolmogorov hypothesis (Kolmogorov, 1962), due to turbulence intermittency $\epsilon$ distribution depends on the scale on which it is evaluated. This dependence inside clouds was investigated experimentally by Siebert et al. (2010).

We added a comment about this issue in sec. 5.3.

In contrast to the PROFs, the LEG-derived exponents stay mostly close to 2/3 or -5/3, accordingly, while the correlations are close to one. We suppose that the observed discrepancy results from the combination of horizontal inhomogeneity and intermittency of turbulence. PROF-derived and LEG-derived parameters should not be directly compared because they represent small and large fluid volumes, respectively. Unfortunately, none of the horizontal segments was performed in the SBL.

12. Page 24, Fig. 17 and Fig. 18: I would use the same scale on the y-axis for Fig. 17 and Fig. 18 for better comparison between the coupled and decoupled case. Furthermore, I would suggest adding the sublayer in brackets next to the altitude.

The figures were modified according to the suggestions.

13. Page 25, line 506: "Lengthscales" should be changed to "Length scales" throughout the paper.

The text was corrected accordingly.

14. Page 25, line 512 and line 524: One of the "$\lambda_u$" in the ratio should be replaced by "$\lambda_w$".

Obviously. We are sorry for the typo.

[Figure]

**Figure 17 corrected.** Spectral anisotropy ratio in the coupled STBL (flight #5). The horizontal dotted line denotes the 4/3 level expected for isotropy in inertial range.

**Figure 18 corrected.** As in Fig. 17 but for the decoupled STBL (flight #14).

15. Page 32, line 669: Change "imortant" to "important"

    Corrected.

16. Page 32, line 670: Change "proprties" to "properties"

    Corrected.

**References**

Betts, A. K.: Non-precipitating cumulus convection and its parameterization, Quarterly Journal of the Royal Meteorological Society, 99, 178–196, https://doi.org/10.1002/qj.49709941915, 1973.

Frehlich, R., Meillier, Y., Jensen, M. L., and Balsley, B.: A statistical description of small-scale turbulence in the low-level nocturnal jet, Journal of the Atmospheric Sciences, 61, 1079–1085, https://doi.org/10.1175/1520-0469(2004)061<1079:ASDOST>2.0.CO;2, 2004.

Jones, C. R., Bretherton, C. S., and Leon, D.: Coupled vs. decoupled boundary layers in VOCALS-REx, Atmospheric Chemistry and Physics, 11, 7143–7153, https://doi.org/10.5194/acp-11-7143-2011, 2011.

Katzwinkel, J., Siebert, H., and Shaw, R. A.: Observation of a Self-Limiting, Shear-Induced Turbulent Inversion Layer Above Marine Stratocumulus, Boundary-Layer Meteorology, 145, 131–143, https://doi.org/10.1007/s10546-011-9683-4, https://link.springer.com/article/10.1007/s10546-011-9683-4, 2012.

Kolmogorov, A. N.: A refinement of previous hypotheses concerning the local structure of turbulence in a viscous incompressible fluid at high Reynolds number, Journal of Fluid Mechanics, 13, 82–85, https://doi.org/10.1017/S0022112062000518, 1962.

Lampert, A., Hartmann, J., Pätzold, F., Lobitz, L., Hecker, P., Kohnert, K., Larmanou, E., Serafimovich, A., and Sachs, T.: Comparison of Lyman-alpha and LI-COR infrared hygrometers for airborne measurement of turbulent fluctuations of water vapour, Atmospheric Measurement Techniques, 11, 2523–2536, https://doi.org/10.5194/amt-11-2523-2018, 2018.

Lenschow, D. H., Mann, J., and Kristensen, L.: How Long is Long Enough when Measuring Fluxes and Other Turbulence Statistics?, Tech. rep., University Corporation for Atmospheric Research, 1993.

Lenschow, D. H., Mann, J., and Kristensen, L.: How long is long enough when measuring fluxes and other turbulence statistics?, Journal of Atmospheric and Oceanic Technology, 11, 661–673, https://doi.org/10.1175/1520-0426(1994)011<0661:HLILEW>2.0.CO;2, 1994.

Muschinski, A., Frehlich, R. G., and Balsley, B. B.: Small-scale and large-scale intermittency in the nocturnal boundary layer and the residual layer, Journal of Fluid Mechanics, 515, 319–351, https://doi.org/10.1017/S0022112004000412, 2004.

Siebert, H. and Muschinski, A.: Relevance of a tuning-fork effect for temperature measurements with the Gill solent HS ultrasonic anemometer-thermometer, Journal of Atmospheric and Oceanic Technology, 18, 1367–1376, https://doi.org/10.1175/1520-0426(2001)018<1367:ROATFE>2.0.CO;2, 2001.

Siebert, H., Wendisch, M., Conrath, T., Teichmann, U., and Heintzenberg, J.: A new tethered balloon-borne payload for fine-scale observations in the cloudy boundary layer, Boundary-Layer Meteorology, 106, 461–482, https://doi.org/10.1023/A:1021242305810, 2003.

Siebert, H., Franke, H., Lehmann, K., Maser, R., Saw, E. W., Schell, D., Shaw, R. A., and Wendisch, M.: Probing finescale dynamics and microphysics of clouds with helicopter-borne measurements, Bulletin of the American Meteorological Society, 87, 1727–1738, https://doi.org/10.1175/BAMS-87-12-1727, 2006a.

Siebert, H., Lehmann, K., and Wendisch, M.: Observations of small-scale turbulence and energy dissipation rates in the cloudy boundary layer, Journal of the Atmospheric Sciences, 63, 1451–1466, https://doi.org/10.1175/JAS3687.1, 2006b.

Siebert, H., Shaw, R. A., and Warhaft, Z.: Statistics of small-scale velocity fluctuations and internal intermittency in marine stratocumulus clouds, Journal of the Atmospheric Sciences, 67, 262–273, https://doi.org/10.1175/2009JAS3200.1, 2010.

Siebert, H., Szodry, K.-E., Egerer, U., Wehner, B., Henning, S., Chevalier, K., Lückerath, J., Welz, O., Weinhold, K., Lauermann, F., Gottschalk, M., Ehrlich, A., Wendisch, M., Fialho, P., Roberts, G., Allwayin, N., Schum, S., Shaw, R. A., Mazzoleni, C., Mazzoleni, L., Nowak, J. L., Malinowski, S. P., Karpinska, K., Kumala, W., Czyzewska, D., Luke, E. P., Kollias, P., Wood, R., and Mellado, J. P.: Observations of Aerosol, Cloud, Turbulence, and Radiation Properties at the Top of the Marine Boundary Layer over the Eastern North Atlantic Ocean: The ACORES Campaign, Bulletin of the American Meteorological Society, 102, E123–E147, https://doi.org/10.1175/bams-d-19-0191.1, 2021.

Wendisch, M., Garrett, T. J., and Strapp, J. W.: Wind tunnel tests of the airborne PVM-100A response to large droplets, Journal of Atmospheric and Oceanic Technology, 19, 1577–1584, https://doi.org/10.1175/1520-0426(2002)019<1577:WTTOTA>2.0.CO;2, 2002.

Yin, B. and Albrecht, B. A.: Spatial variability of atmospheric boundary layer structure over the eastern equatorial Pacific, Journal of Climate, 13, 1574–1592, https://doi.org/10.1175/1520-0442(2000)013<1574:SVOABL>2.0.CO;2, 2000.

---

## Author Comment (AC4)

**Authors' Response to the Review by Ian Brooks**

Jakub L. Nowak, Holger Siebert, Kai-Erik Szodry, Szymon P. Malinowski

We are grateful to Ian Brooks for the insightful comments and suggestions on our manuscript. We respond to them in detail below. The original review is given in black, our answers in blue. The responses also mention the specific corrections which were applied to the manuscript.

The results are, for the most part, routine – such boundary layers are well studied (even if our understanding of all the interacting processes is incomplete), and most of the results are in broad agreement with previous studies (as noted in the conclusions). They remain, however, a useful contribution to the field, and do include some unique results – those of very small-scale turbulent properties: profiles of dissipation rate, and isotropy.

We agree with the general assessment that part of our analysis confirms known findings. From our point of view, these analyses are nevertheless important at this point in order to be able to correctly classify and evaluate the observations of small-scale turbulence. This makes the manuscript a bit longer, but we think - like another reviewer - that this is justified in this case. We believe that a reader should be able to navigate through the text using the section and subsection titles, e.g. skip some of the analyses without losing the thread and find the information which is relevant for him.

There is a limit to how much can be gained from analysis of individual case studies. I would encourage the authors to consider expanding their analysis in future to include all the flights from this campaign (many more than the two used here) to produce a more general synthesis of turbulent behaviour for the coupled and decoupled boundary layers.

This is definitely a suggestion that will be considered for the future. Here, we had to find a compromise between detail and scope, so we first decided to focus on two case studies with a lot of detailed analysis. For the future, we will explore individual aspects with more data to make the results more statistically significant.

On the other hand, the ACORES data alone might not be sufficient to provide statistically sound conclusions and we consider extending the analyzed dataset with the available data from other field experiments. In total, there were 17 research flights during the ACORES (Siebert et al., 2021, Table 5). Five of them correspond to clear-sky conditions, four to already dissipating or not yet developed stratocumulus clouds which limits the true STBL observations to 8 flights. Each flight lasted up to about 2 hours. This flight time was always disposed between sampling the cloud top structure and the boundary layer itself.

**Specific comments**

1. The overall structure of the manuscript follows the conventional pattern of background / methods / results / conclusions. This is fine, but I found that the sheer number of different variables being defined resulted in a very long methods sections, where it wasn't always clear what the real utility of a particular parameter was. By the time the reader (or this reader anyway) gets to the relevant results, they've forgotten what all the different symbols and parameters are. It might be worth considering modifying the structure to mix parameter definitions and results – defining/explaining particular quantities immediately prior to presenting the results on them. This is very much a decision to be made on personal preference regarding the readability, I'm sure another reviewer would argue against doing this.

   We did consider the suggested structure of the manuscript. Actually, our first internal draft followed exactly this approach. However, we came to the conclusion that the multiple explanations of the applied methods disturb a consistent presentation of the results. Another argument in favor of a classical structure is that the reader knows where to expect which content. We are aware that our manuscript is rather long and with its structure we intended to enable readers to easily find the information which is the most relevant for them. Some might be interested in the results only, some in the very details of our methods.

2. On a related note, there are a LOT of acronyms defined here, not all of them are used very often (eg 'CB' is only used 6 times after being defined... not worth the space saving traded off against having to go back and find out what it means'. I found it easy to confuse many of these because of minor inconsistencies in how the layer names mapped to acronyms– I kept reading 'SCL' as 'sub-cloud layer' instead of 'stratocumulus layer', whereas 'SBL' (sub-cloud layer) I wanted to read as 'stable boundary layer'... which is a common usage, but irrelevant here.

   Following the suggestion of this and other reviewers we reduced the number of acronyms by replacing them with the corresponding expanded expressions, in particular those which were not used frequently in the text. This group includes: SC, BL, ENA, TAS, CTEI, J11, WB04, YA00, CP, DCP. The last two were shortened to C and D, respectively, and only kept in sec. 6 to order the list of conclusions. We prefer to keep the acronyms of the following types:

   – denoting the sublayers of the atmosphere: STBL, SML, TSL, SBL, SCL, EIL, FT, because they are used very frequently in the text as well as in tables and figures,

   – denoting our flight segments: PROF, LEG, for the same reason,

   – commonly used abbreviations: TKE, LCL, SFC, PSD, because we expect them to be familiar to the readers,

   – names of instruments or platforms: ACTOS, SMART-HELIOS, MODIS, GPS, for the same reason.

   Moreover, we added the expanded names of the sublayers to the headings of Tables A1 and A2.

3. Figure 2 and 4 – it might be useful to indicate cloud base and top on the figures so the reader can immediately see how the flight legs relate to cloud level. The line style for different sections of the flight track are consistent with those used on the later profile plots - this is clear for fig 4 (flight 14) where the profile plots show 3 distinct profiles; but less so on

fig 2 (flight 5) where there are only 2 line types. It appears that in the profile plots the dashed line, which looks like a single deep profile, is actually a composite of several profile sections separated in time, and spanning different altitude ranges. This is fine, but should be made explicit since it has a bearing on variability of the data.

The figures were modified. The individual penetrations (determined manually based on $\theta_l$, $q_v$ and $q_l$ records) of the EIL top, SCL top (=EIL base), SCL base (=SBL top), TSL top (=SBL bottom) and TSL base were indicated in the altitude profile with additional symbols. The labels denoting PROFs and LEGs were added at the top. The grey line illustrating altitude profile for the whole flight was plotted only outside the colored segments to improve the visibility of the black line used within the segments.

Indeed, the same line style was used for several PROFs which do not overlap in their altitude ranges. We agree this might have been confusing. Therefore, additional short explanation was added to the text and the captions.

[Figure]

**Figure 2 corrected.** ACTOS altitude in flight #5 with marked selected profiles and horizontal legs. PROFs are ordered chronologically, LEGs are ordered according to their altitude. Line styles of the PROFs are consistent with the figures in following sections Altitude ranges corresponding to PROF2-PROF5 of this flight do not overlap and are all marked with dotted lines. Dots indicate the penetrations of the boundaries of the specific sublayers described in sec. 3.

[Figure]

**Figure 4 corrected.** As in Fig. 2 but for flight #14. Line styles of the PROFs are consistent with the figures in following sections. PROF1-PROF3 are all marked with dotted lines because their altitude ranges do not overlap.

4. At various points in the discussion of results, specifically the plots of profiles and leg-averaged values, reference is made to a particular flight leg 'LEG2', 'LEG3' etc. I found this unhelpful, since I couldn't immediately identify which leg was which on the plots... what altitude was it? It would be more useful to simply refer to the altitude of the leg.

The legs can be identified by referring back to figures 2 and 4, but (a) that requires the reader to go searching back for the relevant figure, and (b) there is a potential cause for confusion, because the leg numbering (assuming it is chronological. . . this is never explicitly stated) appears to be inconsistent when referred to the profiles, since for flight 5 the legs start high and word down, and on flight 14 start low and work up (and then down again for final leg). All we really need to know in the discussion is the altitude, the leg number is a distraction'

The numbering of LEGs and PROFs in our manuscript was chronological which was stated in line 111, page 4. We agree that such a convention may be confusing for the reader. Therefore, the ordering of LEGs was changed into LEGX where X stands for mean altitude (m a.s.l.). For PROFs, we kept the chronological numbering. We improved the clarity of the relevant explanation in sec. 2.3. In addition, the labels denoting PROFs and LEGs were added at the top of Figs. 2 and 4.

Segments of two types were selected from the measurement records: profiles (PROFs) and horizontal legs (LEGs). For convenience, for each flight PROFs are ordered chronologically according to their time of execution and referred to as PROF1-PROF5 while LEGs are ordered according to their mean altitude (m above sea level).

5. Line 169: 'Negative values suggest instability. . . ' – for clarity it would be useful to explicitly state the variables involved here 'Negative values of Dql suggest. . . '

Corrected accordingly. 'Negative values of $\Delta\theta_l$ and $\Delta z$ suggest instability ... '

6. Line 171 'The parameter of YA00. . . ' – again, be clear and name the parameter, not (just) the paper where it was first defined. . . make it easy on the reader.

Corrected accordingly. 'The parameter $\mu$ is plotted ...'

7. Line 186: 'probably there were some clearings. . . ' – while the effects of such clear air regions will get averaged out by the vertical binning/averaging/smoothing applied to generate the profiles, it ought to be possible to identify if they actually occur from the raw, high rate data, and not have to rely on a vague statement of 'probably'.

We verified there were indeed some cloud clearings indicated by our high rate data of $q_l$. See the plot below. The text was updated accordingly.

There were cloud clearings penetrated as ACTOS moved along the slanted path, visible in the high rate records of $q_l$ (not shown here).

[Figure]

**Figure.** Cloud penetration during PROF1 of flight #5.

8. Line 202-203: 'Suitable normalisation...' – Purely my preference, but I'd cut this line. I don't think it adds anything useful unless you go into detail about the normalisation & averaging referred to.

We removed the sentence. By suitable normalization we meant the method of Ghate et al. (2015). They identified STBL sublayers in each sounding, normalized the height so that each sublayer has the depth of 1 and then averaged the relevant properties at the normalized heights. This method of profile averaging preserves the sublayer structure despite the depths of the sublayers may vary between the individual soundings.

9. Line 232: what are the instrument issues that resulted in problems with the lateral wind components? It's not essential to document this, but, depending on the cause, might be useful for other researchers trying to make similar measurements.

We observed artificial jumps of the amplitude 0.1 - 1 $\mathrm{m\,s^{-1}}$ in the records of the lateral horizontal component which seems to appear for true air speeds above about 12 $\mathrm{m\,s^{-1}}$ or so. There were virtually no signs of such behavior in the records of the other two velocity components. As far as we know, the issue is specific for that ultrasonic anemometer model. Wind tunnel investigations suggest a problem with the transformation between the axis aligned coordinate system into an orthogonal system. The artifacts were almost not visible along the transducer pairs. However, you have to make a decision which system you want to select for data storage before the flight - we stored in an orthogonal system. Nonetheless, we are not absolutely sure about the reason. Therefore, we prefer not to share the speculations in the published paper until a proper investigation is completed. We added short information to the text:

The lateral channel of the ultrasonic anemometer was affected by a substantial level of artificial fluctuations (up to 1 $\mathrm{m\,s^{-1}}$ in amplitude) due to instrumental issues. The origin of this problem is under investigation. It seems to appear for true air speed above about 12 $\mathrm{m\,s^{-1}}$ which makes it relevant for most of the flight time.

10. Line 294 and 307: both reference a 'lateral component' when the parameter referred to is derived from vertical velocity. Yes, w is 'lateral' with respect to the mean wind vector, but it might be clearer here to be explicit and refer to the 'vertical component', not least because you have previously noted problems with the 'lateral' velocity measurements, where lateral refers to the horizontal cross-wind component, and so is a potential source of confusion.

We are grateful for pointing out this inconsistency. Indeed, the reason for the confusion was that we used the word 'lateral' to name two different aspects: (1) the component $v$ and (2) the orientation with respect to the mean wind vector which is relevant for turbulence theory and the choice of the constant $C$ (then both $v$ and $w$ are considered lateral in contrast to longitudinal $u$). We corrected the description according to the suggestion by sticking to the meaning (1).

11. Line 374-377: The unexpectedly high variances above cloud are presumed to be artefacts resulting from the presence of gravity waves. While I agree that is quite likely, it should be possible to demonstrate it. Coherence/phase/amplitude plots of the correlation between vertical velocity and the other variables should show a clear scale of waves. Power spectra or ogive plots of variances/covariances should also show that most of the variance/covariance results from a narrow range of wavelengths that can be related to gravity waves.

We prepared a brief analysis of the fluctuations recorded in the first half of LEG1079 for which we noticed some quasi-regular oscillations. The figures below present power spectra of $w$, $u$, $q_v$, $\theta_v$ and cospectra of $wu$, $wq_v$, $w\theta_v$. We applied Welch scheme implemented in MATLAB functions `pwelch` and `cpsd`.

[Figure]

[Figure]

**Figure.** Power spectrum density of the first half of LEG1079 normalized by its maximum value (linear scale).

**Figure.** Cospectrum for the correlations in the first half of LEG1079 normalized by its maximum absolute value.

All the power spectra exhibit a pronounced peak at the wavelength of about 450 m. Hence, most of the variance can be attributed to such oscillations. The same can be observed in the cospectra. Thus, most of the covariance can be attributed to those range of wavelengths. We added a short comment justifying our speculations to the manuscript.

Estimated values of the TKE are also large in the FT above the temperature inversion. This is rather an artifact due to the presence of gravity waves favored under stable conditions (the power spectra of $w$, $u$, $q_v$, $\theta_v$ and the cospectra of $wu$, $wq_v$ and $w\theta_v$ indicate the dominant contribution of the wavelength of about 450 m). Recall that LEG1079 was flown very close to the EIL and the cloud top which often features undulated interface.

12. Line 390: The statement regarding T and q as being passive tracers with no significant sources at the transition layer is... arguable. There is no real 'source', but for the SML, the gradient across decoupling transition layer acts as a source/sink term, entrainment mixing brings drier/warmer air down to top of SML (local effective source/sink). There must be some mixing to give high T/Q variances here.

Then...' The TSL features the gradient of qv (c.f. Fig. 6) which might explain increased local fluctuations.' – what other source of increased fluctuations could there be here?

This is a misunderstanding due to the unclear formulation of our point. We just do not consider gradient production (Term IV of the variance budget equations in the formulation of Stull (1988), his Eqs. (4.3.2) and (4.3.3)), as 'sources'.

By 'sources' we meant the processes of the type described by Term VIII in Stull's Eq. (4.3.3), e.g. radiative cooling. Obviously, gradient production is present in our case. The sentence was rewritten to clarify this issue:

> $T$ and $q_v$ can be considered passive scalars which undergo mixing. The increased variances are caused by gradient production (Term IV in the variance budget equations in the formulation of Stull (1988), his Eqs. (4.3.2) and (4.3.3)) rather than by any diabatic sources.

13. Line 420: You 'speculate that the drivers of convection, i.e. radiative and evaporative cooling, are not efficient in this situation'. What is different about 'this situation' that either of these processes should be different? You can evaluate the evaporative cooling and CTEI parameter... is this weaker than for the other case? Certainly the latent and sensible heat fluxes are much smaller in cloud here than for flight 5.

Radiative cooling is more difficult to assess without direct measurements of the radiative fluxes, but there may be clues available. You mention the availability of the ARM remote sensing data... does that show a higher cloud deck that might reduce the radiative cooling from cloud top? This case does have a slightly thicker cloud and so higher LWC at cloud top... this will slightly modify (sharpen) the LW cooling and SW heating profiles, and maybe shift the relative positions of their peaks in the vertical, changing the balance of heating/cooling.

Our speculative comment resulted from the observation of relatively small $Q_s$, $Q_l$ and $B$ in the cloud (LEG992). This LEG was performed below the cloud top but not exactly at the interface or in the EIL, so the fluxes do not represent the entrainment of warm and dry air from the FT but rather parcels which were cooled in the cloud top region and descend through the cloud (Gerber et al., 2016).

Regarding evaporative cooling, the CTEI parameter $\kappa$ (its estimation was given in sec. 3.2) is indeed smaller for the decoupled STBL (0.34) than for the coupled (0.71), yet exceeds the critical value for buoyancy reversal in both cases. This is consistent with the claim of less efficient evaporative cooling in our decoupled stratocumulus.

Regarding radiative cooling, there is evidence that there was no overlying cloud layer during flight #14. No additional clouds were reported by the scientists onboard the helicopter. There are also no overlying clouds visible in the measurements performed by the Ka-band cloud radar and by the ceilometer at ARM ENA site (see the figures below). Moreover, in the substantial region around the operation area, the satellite products derived from MODIS onboard Aqua (NASA Worldview portal) indicate cloud top temperature in the class of 285-290 K and cloud top height in the class of 800-1600 m, both consistent with our observations of stratocumulus top.

Radiative fluxes were measured in the course of the ACORES campaign by the radiometers onboard ACTOS and SMART-HELIOS. However, the data require validation and careful interpretation (e.g. related to the platform inclination and orientation changing during flight maneuvers). This is subject of ongoing work performed by the Atmospheric Radiation research group at the University of Leipzig.

Following the points raised by the reviewer, we reformulated the argumentation in the manuscript:

[Figure]

**Figure.** ARM ENA Ka-band Zenith Radar reflectivity. Figure downloaded from ARM Plot Browser (https://plot.dmf.arm.gov/plotbrowser/).

[Figure]

**Figure.** ARM ENA Ceilometer normalized backscatter coefficient. Figure downloaded from ARM Plot Browser (https://plot.dmf.arm.gov/plotbrowser/).

> Both sensible and latent heat fluxes observed in the cloud (LEG992) are small, in contrast to the coupled case. Together with rather moderate $B$ in the cloud this suggests that the drivers of convection, i.e. radiative and evaporative cooling, are not as efficient in this situation which might have been one of the reasons why decoupling occurred. Cloud top entrainment instability parameter $\kappa$ (sec. 3.2) is indeed smaller in the decoupled cloud in comparison to the coupled one which implies less efficient evaporative cooling. However, the comparison of radiative cooling effects between the cases requires further investigation.

14. Line 422: 'moisture delivery from the ocean surface to the cloud might be more difficult in the decoupled STBL' – yes, it ought to be much more difficult.

> We changed 'might' into 'ought to'.

15. Line 458: is the departure of measurements from theoretical expectations for homogeneity, isotropy and stationarity here a result of evaluating them from slant profiles? You note the horizontal legs are in much better agreement with theory, suggesting the profile results are not truly representative.

The results on small-scale turbulence, including $\epsilon$, $s$ and $p$, should not be compared between PROFs and LEGs in a straightforward way. They are representative for small and large fluid volumes, respectively. According to the refined Kolmogorov hypothesis (Kolmogorov, 1962), due to turbulence intermittency $\epsilon$ distribution depends on the scale on which it is evaluated. This dependence inside clouds was investigated experimentally by Siebert et al. (2010).

In our opinion, the observed discrepancy between PROF-derived and LEG-derived parameters stems from the combination of horizontal inhomogeneity and intermittency of turbulence. In fact, horizontal segments may cover various air volumes differing in turbulence intensity and its properties, e.g. dissipation rate or inertial range scaling. In the region of decaying turbulence, which is likely in the SCL and the SBL of the decoupled boundary layer, there can be even laminar patches. Under such conditions, the scaling of a power spectrum or structure function is dominated by the most intensive portions. In contrast, PROFs are considered to capture local properties. Note that the climb rate of the helicopter is much higher than of a typical research aircraft (c.f. Siebert et al., 2021, sidebar "Profiling with aircraft and helicopter"), so the averaging in horizontal is rather limited in comparison with typical aircraft data.

We commented on this issue in sec. 5.3.:

> In contrast to the PROFs, the LEG-derived exponents stay mostly close to 2/3 or -5/3, accordingly, while the correlations are close to one. We suppose that the observed discrepancy results from the combination of horizontal inhomogeneity and intermittency of turbulence. PROF-derived and LEG-derived parameters should not be directly compared because they represent small and large fluid volumes, respectively. Unfortunately, none of the horizontal segments was performed in the SBL.

16. Line 555: 'which suggests important contribution of moisture to buoyancy' – I agree, but this could be evaluated properly. Buoyancy flux (virtual potential temperature flux) can be broken down into the sensible and latent heat contributions and their ratio determined.

We evaluated the terms contributing to virtual potential temperature flux using the approximation valid under dry conditions (e.g De Roode and Duynkerke, 1997):

$$\langle w'\theta'_v \rangle = \langle w'\theta' \rangle + \theta\varepsilon\langle w'q'_v \rangle \tag{1}$$

where $\varepsilon = R_v/R_d - 1$. The results listed in the table below confirm that the moisture term plays a significant role in the lower part of the boundary layer.

The information about this fact was added to sec. 5.2 and sec.6.

> At low levels in the atmosphere (at the surface and in LEG307) the contribution of moisture transport to buoyancy is of the same order as the contribution of heat transport (not shown).

> In the lower part of the STBL (at the surface, in LEG143 and LEG287) the contribution of moisture transport to buoyancy is of the same order as the contribution of heat transport (not shown).

**Table 1.** Contributions to virtual potential temperature flux inside the boundary layer. Left column for each term is a LEG-derived value, right column is the variability among subsegments.

| | Height [m] | $\langle w'\theta'\rangle$ $[10^{-3}\,\mathrm{mK\,s^{-1}}]$ | | $\varepsilon\theta\langle w'q_v'\rangle$ $[10^{-3}\,\mathrm{mK\,s^{-1}}]$ | |
|---|---|---|---|---|---|
| **Flight #5** | 0 | 9.0 | - | 7.6 | - |
| | 307 | 3.4 | 2.2 | 3.0 | 1.4 |
| | 553 | 6.1 | 5.6 | 0.3 | 1.4 |
| | 819 | 33.1 | 17.8 | 6.5 | 4.8 |
| **Flight #14** | 0 | 5.5 | - | 6.3 | - |
| | 143 | 7.4 | 2.7 | 4.6 | 2.1 |
| | 287 | 1.4 | 3.4 | 2.4 | 1.7 |
| | 448 | 3.1 | 2.3 | 0.4 | 2.7 |
| | 992 | 2.2 | 3.4 | 0.7 | 1.5 |

In both cases, latent heat flux qualitatively resembles the profile of $B$ which is consistent with the considerable contribution of moisture transport to buoyancy in the lower part of the STBL.

**Minor issues (grammar, typos, etc)**

While overall, the manuscript is clear and well written, there are many minor grammatical issues – notably missing definitive articles: '...in the cloud top region...', '...in the inertial subrange...' etc. I have noted all those that jumped out at me below, but I'm sure I've missed more.

We are impressed by the language editing provided by the reviewer. As we are not native speakers, we rely on him and applied all the minor corrections suggested.

1. Line 4: '...in cloud top region' -> '...in the cloud top region'

   Corrected.

2. Line 12: 'in inertial subrange' -> 'in the inertial subrange'

   Corrected.

3. Line 22: 'They occupy..., preferably in the conditions of large-scale subsidence.' – 'preferably' is the wrong word (implies an ideal choice or active preference', 'preferentially' is closer to the meaning required (with greater likelihood)

Changed to 'preferentially'.

4. Line 28: 'Primary mechanism...' -> 'The primary mechanism...'

Corrected.

5. Line 29: 'Additional source of turbulence...' -> ' An additional source of turbulence...'

Corrected.

6. Line 32: '...dependent on the level in which SC is coupled...' – 'in' isn't the right word here, and the meaning intended isn't entirely clear, either '...dependent on the level at which SC is coupled...' (if the issue of concern is the altitude at which decoupling occurs) or '...dependent on the level to which SC is coupled...' (if the issue is whether, or how strongly decoupled the BL is).

Changed to 'to which'.

7. Line 35: '...structure features adiabatic lapse rate (dry below cloud, moist inside), strong capping inversion at the top, near-constant concentration of moist-conserved variables...' -> '...structure features an adiabatic lapse rate (dry below cloud, moist inside), a strong capping inversion at the top, and near-constant concentration of moist-conserved variables...'

Corrected.

8. Line 40: ' Stable or...' -> 'A stable or...'

Corrected.

9. Line 62: '...in conventional rationale...' -> '...in the conventional rationale...'

Corrected.

10. Line 105: '...depended on local cloud...' -> '...depended on the local cloud...'

Corrected.

11. Line 106: 'usual strategy involved:...' -> 'the usual strategy involved...'

Corrected.

12. Line 109: '...and flight pattern...' -> '...and a flight pattern...'

Corrected.

13. Line 140: 'Brunt-Vaisala frequency...' -> 'The Brunt-Väisälä frequency...'

Corrected.

14. Line 143: '...quantifies vertical gradient...' -> '...quantifies the vertical gradient...'

    Corrected.

15. Line 166: '...as BL mean.' -> '...as the BL mean.'

    Corrected.

16. Line 183: '...where it features the increase of...' -> '...where it features an increase of...'

    Corrected.

17. And '...analogously, with the decrease of...' -> '...analogously, with a decrease of...'

    Corrected.

18. Line 184-185: '...capped by the layer of...' -> '...capped by a layer of...'

    Corrected.

19. Line 239: 'Described modification...' -> 'The modification described...'

    Corrected.

20. Line 243: '...with simple...' -> '...with a simple...'

    Corrected.

21. Line 244: '...from original signal.' -> '...from the original signal.'

    Corrected.

22. Line 249: '...taking average along LEG...' -> '...taking the average along the leg...'

    Corrected.

23. Line 252: 'Worth to remember...' -> 'It is worth remembering...'

    Corrected.

24. Line 260: '...such approach...' -> '...such an approach...'

    Corrected.

25. Line 266: 'Range of scales...' -> 'The range of scales...'

    Corrected.

26. Line 266: '...limited by the smaller among spatial resolutions of two multiplied signals...' the word smaller here might be read as implying the smaller scale (ie, higher resolution), suggest -> '...limited by the lowest spatial resolution of the two multiplied signals...'

    Corrected.

27. Line 267-268: following the previous statement, you note the scales of individual measurements, but it would be helpful to be explicit and state the resulting scale for the final fluxes.

Sentence added: 'As a result, $\langle w'\theta'\rangle$, $\langle w'u'\rangle$ and $\langle w'\theta'\rangle$ are resolved down to $\sim0.5$ m while $\langle w'q_v'\rangle$ down to $\sim1$ m.'

28. Line 317: 'In case of LEGs...' –> 'In the case of LEGs...'

Corrected.

29. Line 338: 'Similar approach...' -> 'A similar approach...'

Corrected.

30. Line 340: 'Such value of...' -> 'Such a value of...'

Corrected.

31. Line 345: 'Integral lengthscale...' -> 'The integral length scale...'

Corrected.

32. Line 347: '...integral of autocorrelation function...' -> '...integral of the autocorrelation function...'

Corrected.

33. '...in formal definition...' -> '...in the formal definition...'

Corrected.

34. Line 353: 'At Taylor microscale...' -> 'At the Taylor microscale...'

Corrected.

35. Line 365-366: The statement 'Depending on flight segment type, they are illustrated with continuous profiles (PROF) and/or dots with errorbars (LEG).' Is redundant, delete.

Ok, removed.

36. Line 372: '...reaches minimum value...' -> '...reaches a minimum value...'

Corrected.

37. Line 387: '...resemble typical mixed layer...' -> '...resemble a typical mixed layer...'

Corrected.

38. Line 390: '...exhibit maximum...' -> '...exhibit a maximum...'

Corrected.

39. Line 398-399: The statement 'while the shear production at the bottom and at the top of the boundary layer' is incomplete... needs some statement about the shear production.

   Corrected to 'while the shear production is expected to be significant at the bottom and at the top of the boundary layer'.

40. Lines 395, 402, 406: statements about results are phrased as 'seems to', 'appears to be' etc. Unless there is real doubt, be definitive... is it as stated or not?

   Corrected. Speculative verbs were removed.

41. Line 495: '... immensely stable...' -> '... strongly stable...' (immensely might be overstating things a bit).

   Corrected.

42. Line 561: '... vanishes at the level...' -> '... vanishes at a level...'

   Corrected.

43. Line 568-569: 'Vertical velocity variance suggests the profile somewhat different than the convective similarity scaling' -> 'The vertical velocity variance suggests a profile somewhat different than the convective similarity scaling'

   Corrected.

44. Line 659: 'Main processes...' -> 'The main processes...'

   Corrected.

45. Line 669: 'imortant' -. 'important'

   Corrected.

46. Line 670: 'relevant systematical...' – '... relevant systematic...'

   Corrected.

**References**

De Roode, S. R. and Duynkerke, P. G.: Observed lagrangian transition of stratocumulus into cumulus during ASTEX: Mean state and turbulence structure, Journal of the Atmospheric Sciences, 54, 2157–2173, https://doi.org/10.1175/1520-0469(1997)054<2157:OLTOSI>2.0.CO;2, 1997.

Gerber, H., Malinowski, S. P., and Jonsson, H.: Evaporative and Radiative Cooling in POST Stratocumulus, Journal of the Atmospheric Sciences, 73, 3877–3884, https://doi.org/10.1175/JAS-D-16-0023.1, 2016.

Ghate, V. P., Miller, M. A., Albrecht, B. A., and Fairall, C. W.: Thermodynamic and radiative structure of stratocumulus-topped boundary layers, Journal of the Atmospheric Sciences, 72, 430–451, https://doi.org/10.1175/JAS-D-13-0313.1, 2015.

Kolmogorov, A. N.: A refinement of previous hypotheses concerning the local structure of turbulence in a viscous incompressible fluid at high Reynolds number, Journal of Fluid Mechanics, 13, 82–85, https://doi.org/10.1017/S0022112062000518, 1962.

Siebert, H., Shaw, R. A., and Warhaft, Z.: Statistics of small-scale velocity fluctuations and internal intermittency in marine stratocumulus clouds, Journal of the Atmospheric Sciences, 67, 262–273, https://doi.org/10.1175/2009JAS3200.1, 2010.

Siebert, H., Szodry, K.-E., Egerer, U., Wehner, B., Henning, S., Chevalier, K., Lückerath, J., Welz, O., Weinhold, K., Lauermann, F., Gottschalk, M., Ehrlich, A., Wendisch, M., Fialho, P., Roberts, G., Allwayin, N., Schum, S., Shaw, R. A., Mazzoleni, C., Mazzoleni, L., Nowak, J. L., Malinowski, S. P., Karpinska, K., Kumala, W., Czyzewska, D., Luke, E. P., Kollias, P., Wood, R., and Mellado, J. P.: Observations of Aerosol, Cloud, Turbulence, and Radiation Properties at the Top of the Marine Boundary Layer over the Eastern North Atlantic Ocean: The ACORES Campaign, Bulletin of the American Meteorological Society, 102, E123–E147, https://doi.org/10.1175/bams-d-19-0191.1, 2021.

Stull, R. B.: An Introduction to Boundary Layer Meteorology, Springer Netherlands, Dordrecht, https://doi.org/10.1007/978-94-009-3027-8, 1988.